# Neural mechanisms underlying the hierarchical construction of perceived aesthetic value

Kiyohito Iigaya [1,2,3] ✉, Sanghyun Yi [1], Iman A. Wahle[1], Sandy Tanwisuth[1], Logan Cross[1,4] & John P. O'Doherty [1] ✉

Little is known about how the brain computes the perceived aesthetic value of complex stimuli such as visual art. Here, we used computational methods in combination with functional neuroimaging to provide evidence that the aesthetic value of a visual stimulus is computed in a hierarchical manner via a weighted integration over both low and high level stimulus features contained in early and late visual cortex, extending into parietal and lateral prefrontal cortices. Feature representations in parietal and lateral prefrontal cortex may in turn be utilized to produce an overall aesthetic value in the medial prefrontal cortex. Such brain-wide computations are not only consistent with a feature-based mechanism for value construction, but also resemble computations performed by a deep convolutional neural network. Our findings thus shed light on the existence of a general neurocomputational mechanism for rapidly and flexibly producing value judgements across an array of complex novel stimuli and situations.

How it is that humans are capable of making aesthetic judgments has long been a focus of enquiry in psychology, more recently gaining a foothold in neuroscience with the emerging field of neuroaesthetics[1–10]. Yet in spite of the long tradition of studying value judgments, we still have a very limited understanding of how people form aesthetic value, let alone of the neural mechanisms underlying this enigmatic process. So far, neuroscience studies into aesthetic judgments have been largely limited to identifying brain regions showing increased activity to stimuli with higher compared to lower aesthetic value (e.g.,[11,12]), leaving it an open question of how the brain *computes* aesthetic value from visual stimuli in the first place. To fill this gap, we approach this problem from a computational neuroscience perspective, by leveraging computational methods to gain insight into the neural computations underlying aesthetic value construction.

Considerable progress has been made toward understanding how the brain represents the value of stimuli in the world. Value signals have been found throughout the brain, but most prominently in the medial prefrontal (mPFC) and adjacent orbitofrontal cortex. Activity has been found in this region tracking the experienced value of reward outcomes, as well as during anticipation of future rewards[11,13–25]. The mPFC, especially its ventral aspects, has been found to correlate with the experienced pleasantness of gustatory, olfactory, music, and visual stimuli including faces, but also visual art[12,26–31]. While much is known about how the brain represents value, much less is known about how those value signals come to be generated by the brain in the first place.

A typical approach to this question in the literature to date is to assume that stimuli acquire value through associative learning, in which the value of a particular stimulus is modulated by being associated with other stimuli with extant (perhaps even innate) value. Seminal work has identified key neural computations responsible for implementing this type of reward-based associative learning in the brain[32–34]. However, the current valuation of an object is not solely dependent on its prior associative history. Even novel stimuli never before seen, can be assigned a value judgment[35]. Moreover, the current

[1]Division of Humanities and Social Sciences, California Institute of Technology, 1200 E California Blvd, Pasadena, CA 91125, USA. [2]Department of Psychiatry, Columbia University Irving Medical Center, New York, NY 10032, USA. [3]Center for Theoretical Neuroscience and Mortimer B. Zuckerman Mind Brain Behavior Institute, Columbia University, New York, NY 10027, USA. [4]Department of Computer Science, Stanford University, Stanford, CA, USA.
✉e-mail: ki2151@columbia.edu; jdoherty@caltech.edu

value of a stimulus depends on one's internal motivational state, as well as the context in which a stimulus is presented. Consequently, the value of an object may be computed on-line in a flexible manner that goes beyond simple associative history.

Previous work in neuroeconomics has hinted at the notion that value can be actively constructed by taking into account different underlying features or attributes of a stimulus. For instance, a t-shirt might have visual and semantic components[36], a food item might vary in its healthfulness and taste or in its nutritive content[37,38], an odor is composed of underlying odor molecules[39]. Furthermore, potential outcomes in many standard economic decision-making problems can be described in terms of the magnitude and probability of those outcomes[40,41]. For a given outcome, these individual features of such potential outcomes can be each weighted so that they are taken into account when making an overall value determination.

Building upon these ideas, we recently proposed that the value of a stimulus, including a piece of art, is actively constructed in a two-step process by first breaking down a stimulus into its constituent features and then by recombining these features in a weighted fashion, to compute an overall subjective value judgment[42,43]. In a recent study we showed that it is possible to demonstrate that this same feature-based value construction process can be used to gain an understanding about how humans might value works of art as well as varieties of online photograph images[42]. Using a combination of computer vision tools and machine learning, we showed that it is possible to predict an individual's subjective aesthetic valuation for a work of art and photography, by segmenting a visual scene into its underlying visual features, and then combining those features together in a weighted fashion.

While this prior work[42] provides empirical support for the applicability of the value construction process for understanding aesthetic valuation, nothing is yet known about whether this approach is actually a plausible description of what might actually be occurring in the brain, a crucial step for validating this model as a biological mechanism.

Establishing how the brain might solve the feature integration process for art is uniquely challenging because of the complexity and diversity of visual art. Even in paintings alone, there are an overwhelmingly broad range of objects, themes, as well as styles that are used across artworks. The brain's value computation mechanism therefore needs to generalize across all of these diverse stimuli, in order to compute the value of them reliably. However, it is not known how the brain can transform heterogeneous, high-dimensional input, into a simple output of an aesthetic judgment.

Here we address these challenges by combining computational modelling with neuroimaging data. Following our prior behavioral evidence, we propose that the brain performs aesthetic value computations for visual art through extracting and integrating visual and abstract features of a visual stimulus. In our linear feature summation model (LFS)[42], the input is first decomposed into various visual features characterizing the color or shape of the whole and segments of the paintings. These features are then transformed into abstract high-level features that also affect value judgement (e.g., how concrete or abstract the painting is). This feature space enables a robust value judgment to be formed for visual stimuli even never before seen, through a simple linear regression over the features. We also recently reported that the features predicting value judgment in the LFS model naturally emerge in a generic deep convolutional neural network (DCNN) model, suggesting a close relationship between these two models[42]. Here we test whether these computational models actually approximate what is going on in the brain. By doing so, we will attempt to link an explicit, interpretable feature-based value computation and a generic DCNN model to actual neural computations. Because our model of value construction is agnostic about the type of object that is being valued, our proposed mechanism has the potential to not only

account for aesthetic value computation but also to value judgments across stimulus domains beyond the domain of aesthetics for art.

## Results

### Linear feature summation (LFS) model predicts human valuation of visual art

We conducted an fMRI study in which we aimed to link two complementary computational models (LFS and DCNN) to neural data as a test of how feature-based value construction might be realized in the brain. Rather than collecting noisy data from a large number of participants with very short scanning times to perform group averaging, here, we engaged in deep fMRI scanning of a smaller group of individuals ($n = 6$), who each completed 1000 trials of our art rating task (ART) each over four days of scanning. This allowed us to test for the representation of the features in each individual participant with sufficient fidelity to perform reliable single subject inference. This well-justified approach essentially treats the individual participant as the replication unit, rather than relying on group-averaged data from participants in different studies[44]. This has been a dominant and highly successful approach in most non-human animal studies (e.g.,[14,18,20,22,24,33,45,46]), as well as in two subfields of human psychology and neuroscience: psychophysics and visual neuroscience, respectively (e.g.,[47,48]).

On each trial, participants were presented with an image of a painting on a computer screen and asked to report how much they liked it on a scale of 0 (not at all) to 3 (very much) (Fig. 1a). Each of the participants rated all of the paintings without repetition (1000 different paintings). The stimulus set consisted of paintings from a broad range of art genres (Fig. 1b)[42].

We recently showed that a simple linear feature summation (LFS) model can predict subjective valuations for visual art both for paintings and photographs drawn from a broad range of scenery, objects, and advertisements[42]. The idea is that the subjective value of an individual painting can be constructed by integrating across features commonly shared across all paintings. For this, each image was decomposed into its fundamental visual and emotional features. These feature values are then integrated linearly, with each participant being assigned a unique set of features weights from which the model constructs a subjective preference (Fig. 1c, d). This model embodies the notion that subjective values are computed in a common feature space, whereby overall subjective value is computed as a weighted linear sum over feature content (Fig. 1e, f).

The LFS model extracts various low-level visual features from an input image using a combination of computer vision methods (e.g.,[49]). This approach computes numerical scores for different aspects of visual content in the image, such as the average hue and brightness of image segments, as well as the entirety of the image itself, as identified by machine learning techniques, e.g., Graph-Cuts[50] (Details of this approach are described in the Methods section; see also[42]).

The LFS model also includes more abstract or "high-level" features that can contribute to valuation. Based on previous literature[51,52], we introduced three features: the image is 'abstract or concrete'[53], 'dynamic or still', 'hot or cold'. These three features are introduced in ref. 52, by taking principal components of features originally introduced in ref. 51. We also included a fourth feature: whether the image evinces a positive or negative emotional valence[42]. Note that image valence is not the same as valuation because even negative valence images can elicit a positive valuation (e.g., Edvard Munch's "The Scream"); moreover we have previously shown that valence is not among the features that account for valuation the best[42]. These high-level features across images were annotated by participants with familiarity and experience in art ($n = 13$) as described in our previous study[42]. We then took the average score over these experts' ratings as the input into the model representing the content of each high-level attribute feature for each image. We have previously shown that it is

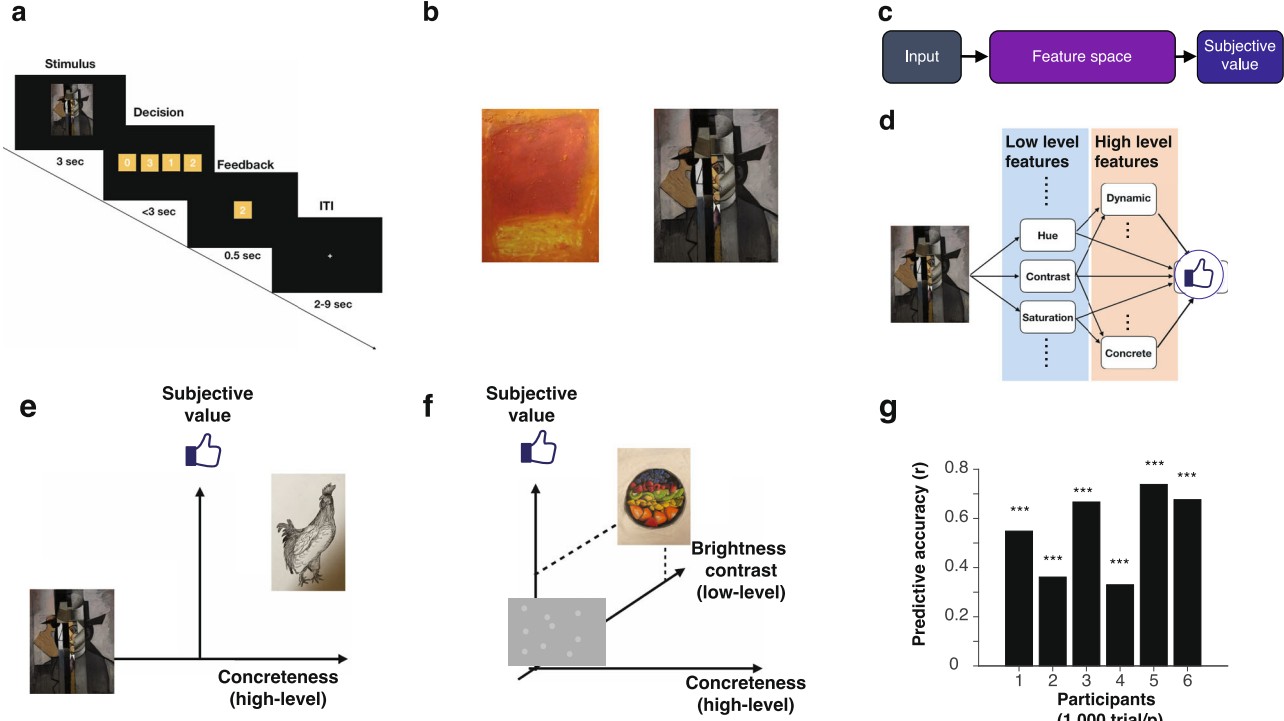

**Fig. 1 | Neuroimaging experiments and the model of value construction.**
**a** Neuroimaging experiments. We administered our task (ART: art rating task) to human participants in an fMRI experiment. Each participant completed 20 scan sessions spread over four separate days (1000 trials in total with no repetition of the same stimuli). On each trial, a participant was presented with a visual art stimulus (paintings) for 3 s. The art stimuli were the same as in our previous behavioral study[42]. After the stimulus presentation, a participant was presented with a set of possible ratings (0,1,2,3), where they had to choose one option within 3 s, followed by brief feedback with their selected rating (0.5 s). The positions of the numbers were randomized across trials, and the order of presented stimuli was randomized across participants. **b** Example stimuli. The images were taken from four categories from Wikiart.org.: Cubism, Impressionism, Abstract art and Color Fields, and supplemented with art stimuli previously used[52]. **c** The idea of value construction. An input is projected into a feature space, in which the subjective value judgment is performed. Importantly, the feature space is shared across stimuli, enabling this mechanism to generalize across a range of stimuli, including novel ones. **d** Schematic of the LFS model[42]. A visual stimulus (e.g., artwork) is decomposed into various low-level visual features (e.g., mean hue, mean contrast), as well as high-level features (e.g., concreteness, dynamics). We hypothesized that in the brain high-level features are constructed from low-level features, and that subjective value is constructed from a linear combination of all low and high-level features. **e** How features can help construct subjective value. In this example, preference was separated by the concreteness feature. Reproduced from[42]. **f** In this example, the value over the concreteness axis was the same for four images; but another feature, in this case, the brightness contrast, could separate preferences over art. Reproduced from[42]. **g** The LFS model successfully predicts participants' liking ratings for the art stimuli. The model was fit to each participant (cross-validated). Statistical significance was determined by a permutation test (one-sided). Three stars indicate $p < 0.001$. Due to copyright considerations, some paintings presented here are not identical to that used in our studies. Credit. Jean Metzinger, Portrait of Albert Gleizes (public domain; RISD Museum).

possible to re-construct significant, if not all of the variance explained by high-level features using combinations of low-level features[42], supporting the possibility that low-level features can be used to construct high level features.

The final output of the LFS model is a linear combination of low- and high-level features. We assumed that weights over the features are fixed for each individual across stimuli, so that we can derive generalizable conclusions about the features used to generate valuation across images. In our behavioral fitting, we treat low-level and high-level features equally as features of a linear regression model, in order to determine the overall predictive power of our LFS model.

In our recent behavioral study, we identified a minimal feature set that predicts subjective ratings across participants using a group-level lasso regression[42]. Here, we applied the same analysis, except that we distinguished low- and high-level features for the purpose of the fMRI analysis. Since our interests hinge on the representational relationship between low- and high-level features in the brain, we first identified a set of low-level features that can predict behavioral liking ratings across participants, further augmenting this to a richer feature set that includes the four human-annotated features. By doing so, we aimed to identify brain signals that are uniquely correlated with low-level and high-level features (i.e., partial correlations between features and fMRI signals).

Before turning to the fMRI data, we first aimed to replicate the behavioral results reported in our previous study[42] in these fMRI participants. Indeed, using the LFS model with the shared feature set, we confirmed that the model could predict subjective art ratings across participants, replicating our previous behavioral findings (Fig. 1g; see Supplementary Figs. 1 and 2 for the estimated weights for each participant and their correlations).

## A deep convolutional neural network (DCNN) model also predicts human liking ratings for visual art

An alternative approach to predict human aesthetic valuation for visual images is to use a generic deep neural network model that takes as its input the visual images, and ordinarily generates outputs related to object recognition. Here, we utilized a standard DCNN (VGG 16[54]) that had been pre-trained for object recognition with ImageNet[55], and adapted it to instead output aesthetic ratings by training it on human aesthetic ratings. This approach means that we do not need to identify or label specific features that lead to aesthetic ratings, instead we can use the network to automatically detect the relevance of an image and use those to predict aesthetic ratings.

Though the nature of computation that DCNN performs is usually very difficult to interpret, we have recently found that this type of

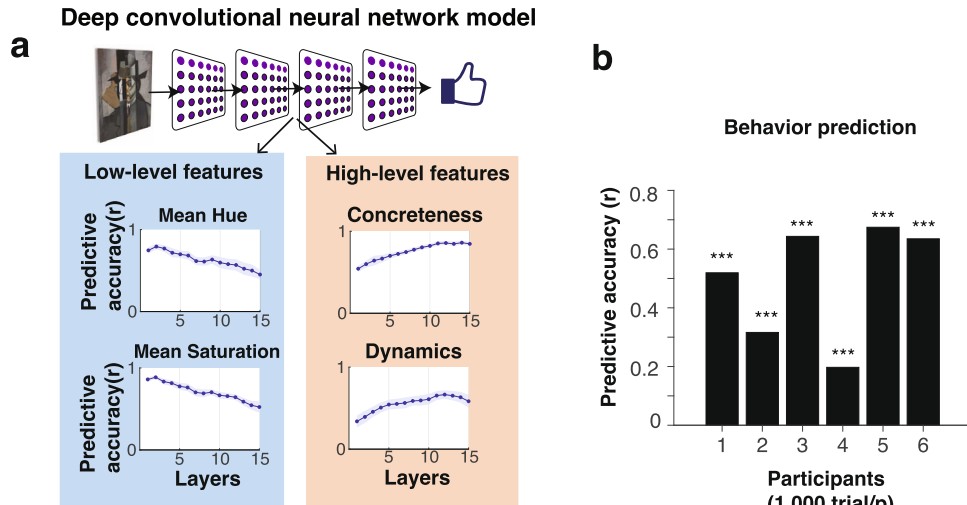

**Fig. 2 | The deep convolutional neural network (DCNN) model naturally encodes low-level and high-level features and predict participants' choice behavior. a** Schematic of the deep convolutional neural network (DCNN) model and the results of decoding analysis[42]. The DCNN model was first trained on ImageNet object classifications, and then the average ratings of art stimuli. We computed correlations between each of the LFS model features and activity patterns in each of the hidden layers of the DCNN model. We found that some low-level visual features exhibit significantly decreasing predictive accuracy over hidden layers (e.g., the mean hue and the mean saturation). We also found that a few

computationally demanding low-level features showed the opposite trend (see the main text). We further found that some high-level visual features exhibit significantly increasing predictive accuracy over hidden layers (e.g., concreteness and dynamics). Results reproduced from[42]. **b** The DCNN model could successfully predict human participants' liking ratings significantly greater than chance across all participants. Statistical significance ($p < 0.001$, indicated by three stars) was determined by a permutation test (one-sided). Credit. Jean Metzinger, Portrait of Albert Gleizes (public domain; RISD Museum).

DCNN model can produce results that are strongly related to the LFS model[42]. In particular, we have found that the LFS model features are represented in the DCNN model, even though we did not train the DCNN on any features explicitly (Fig. 2a). By performing a decoding analysis on each layer of the DCNN, we found that the low-level features show decreased decoding accuracy with increased depth of the layer, while the high-level features are more decodable in deeper layers. This suggests that the DCNN may also utilize similar features to those that we introduced in the LFS model, and the fact that features are represented hierarchically in a DCNN model that is blind to specific features suggests that these features might emerge spontaneously via a natural process of visual and cognitive development through interacting with natural stimuli[56].

Here we found that the DCNN model can predict subjective liking ratings of art across all fMRI participants (Fig. 2b), once again replicating our previous finding in a different dataset[42]. Predictive accuracy across participants was found to be similar to that of the LFS model, though the DCNN model could potentially perform better with even more training.

So far, we have confirmed the validity of our LFS model and the use of a DCNN model to predict human behavioral ratings reported in our new fMRI experiments, replicating our previous behavioral study[42]. Now that we have validated our behavioral predictions, next we turn to the brain data to address how the brain implements the value construction process.

**The subjective value of art is represented in the medial prefrontal cortex (mPFC)**

We first tested for brain regions correlating with the subjective liking ratings of each individual stimulus at the time of stimulus onset. We expected to find evidence for subjective value signals in the medial prefrontal cortex (mPFC), given this is the main area found to correlate with value judgments for many different stimuli from an extensive prior literature, including for visual art (e.g.,[11,12,14,17,23,31,57]). Consistent with our hypothesis, we found that voxels in the mPFC are positively correlated with subjective value across participants (Fig. 3; See

Supplementary Fig. 3 for the timecourse of the BOLD signals in the mPFC cluster). Consistent with previous studies, e.g.,[38,58–63], other regions are also correlated with liking value (Supplementary Figs. 5 and 6).

These subjective value signals could reflect other psychological processes such as attention. Therefore we performed a control analysis with the same GLM with additional regressors that can act as proxies for the effects of attention and memorability of stimuli, operationalized by reaction times, squared reaction times and the deviation from the mean rating[64]. We found that subjective value signals in all participants that we report in Fig. 3c survived this control analysis (Supplementary Fig. 7).

**Visual stream shows hierarchical, graded, representations of low-level and high-level features**

As illustrated in Fig. 1d, and reflecting our hypothesis regarding the encoding of low vs. high-level features across layers of the DCNN, we hypothesized that the brain would decompose visual input similarly, with early visual regions first representing low-level features, and with downstream regions representing high-level features. Specifically, we analyzed visual cortical regions in the ventral and dorsal visual stream[65] to test the degree to which low-level and high-level features are encoded in a graded, hierarchical manner. In pursuit of this, we constructed a GLM that included the shared feature time-locked to stimulus onset. We identified voxels that are significantly modulated by at least one low-level feature by performing an F-test over the low-level feature beta estimates, repeating the same analysis with high-level features. We then compared the proportion of voxels that were significantly correlated with low-level features vs. high-level features in each region of interest in both the ventral and dorsal visual streams. This method allowed us to compare results across regions while controlling for different signal-to-noise ratios in the BOLD signal across different brain regions[66]. Regions of interest were independently identified by means of a detailed probabilistic visual topographical map[65]. Consistent with our hypothesis, our findings suggest that low- and high-level features relevant for aesthetic valuation are indeed

**Fig. 3 | Subjective value (i.e., liking rating).** Subjective value for art stimuli at the time of stimulus onset was found in the medial prefrontal cortex in all six fMRI participants (One-sided *t*-test. An adjustment was made for multiple comparisons: whole-brain cFWE *p* < 0.05 with height threshold at *p* < 0.001).

represented in the visual stream in a graded hierarchical manner. Namely, the relative encoding of high-level features with respect to low-level features dramatically increases across the visual ventral stream (Fig. 4a). We found a similar, hierarchical organization in the dorsolateral visual stream (Fig. 4b), albeit less clearly demarcated than in the ventral case. We also confirmed in a supplementary analysis that referring to feature levels (high or low) according to our DCNN analysis, i.e., by using the slopes of our decoding results[42], did not change the results of our fMRI analyses qualitatively and does not affect our conclusions (see Supplementary Fig. 8).

We also performed additional encoding analysis using cross validation at each voxel of each participant[67]. Specifically, we performed a lasso regression at each voxel with the low- and high-level features that we considered in our original analyses. Hyperparameters are optimized in 12-fold cross validation at each voxel across stimuli. As a robustness check, we determined if our GLM results can be reproduced using the lasso regression analysis. We analyzed how low-level feature weights and high-level feature weights changed across ROIs. For this, we computed the sum of squares of low-level feature weights and the sum of squares of high-level feature weights at each voxel. Because these weights estimates include those that can be obtained by chance, we also computed the same quantities by performing the lasso regression with shuffled stimuli labels (labels were shuffled at every regression). The null distribution of feature magnitudes (the sum of squares) was estimated for low-level features and high-level features at each ROI. For each voxel, we asked if estimated low-level features and high-level features are significantly larger than what is expected from noise, by comparing the magnitude of weights against the weights from null distribution (*p* < 0.001). We then examined how encoding of low-level vs high-level features varied across ROIs, as we did in our original GLM analysis. As seen in Supplementary Fig. 9, the original GLM analysis results were largely reproduced in the lasso regression. Namely, low-level features are more prominently encoded in early visual regions, while high-level features are more prominently encoded in higher visual regions. In this additional analysis, such effects were clearly seen across five out of six participants, while one participant (P1) showed less clear early vs late region-specific differentiation with regard to low vs high-level feature representation. We also note that the model's predictive accuracy in visual regions was lower for this

participant (P1) than for the rest of the participants (Supplementary Fig. 10).

## Non-linear feature representations

We found that features of the LFS model are represented across brain region and contribute to value computation. However, it is possible that nonlinear combinations of these features are also represented in the brain and that these may contribute to value computation. To explore this possibility, we constructed a new set of nonlinear features by multiplying pairs of the LFS model's features (interaction terms). We grouped these new features into three groups: interactions between pairs of low-level features (low-level × low-level), interactions between pairs of low-level and high-level features (low-level × high-level), and interactions between pairs of high-level features (high-level × high-level). To control the dimensionality of the new feature groups, we performed principal component analysis within each of the three groups of non-linear features, and took the first five PCs to match the number of the high-level features specified in our original LFS model. We performed a LASSO regression analysis with these new features and the original features.

We found that in most participants, non-linear features created from pairs of high-level features produced significant correlations with neural activity across multiple regions, while also showing similar evidence for a hierarchical organization from early to higher-order regions, as found for the linear high-level features (Fig. 5, Supplementary Fig. 11). Though comparisons between separately optimized lasso regressions should be cautiously interpreted, the mean correlations of the model with both linear and nonlinear features across ROIs showed a slight improvement in predictive accuracy compared to the original LFS model with only linear features (Supplementary Fig. 10), while the DCNN model features out-performed both the original LFS model and the LFS model + nonlinear features.

Indeed, nonlinear features created from pairs of high-level features significantly contribute more to behavioral choice predictions than do other nonlinear features not built solely from high-level features (Supplementary Fig. 12). The first principal component of high level x high level features well captured three participants (3, 5, 6) behavior, while other participants show somewhat different weight profiles. However, we found that these newly added features only

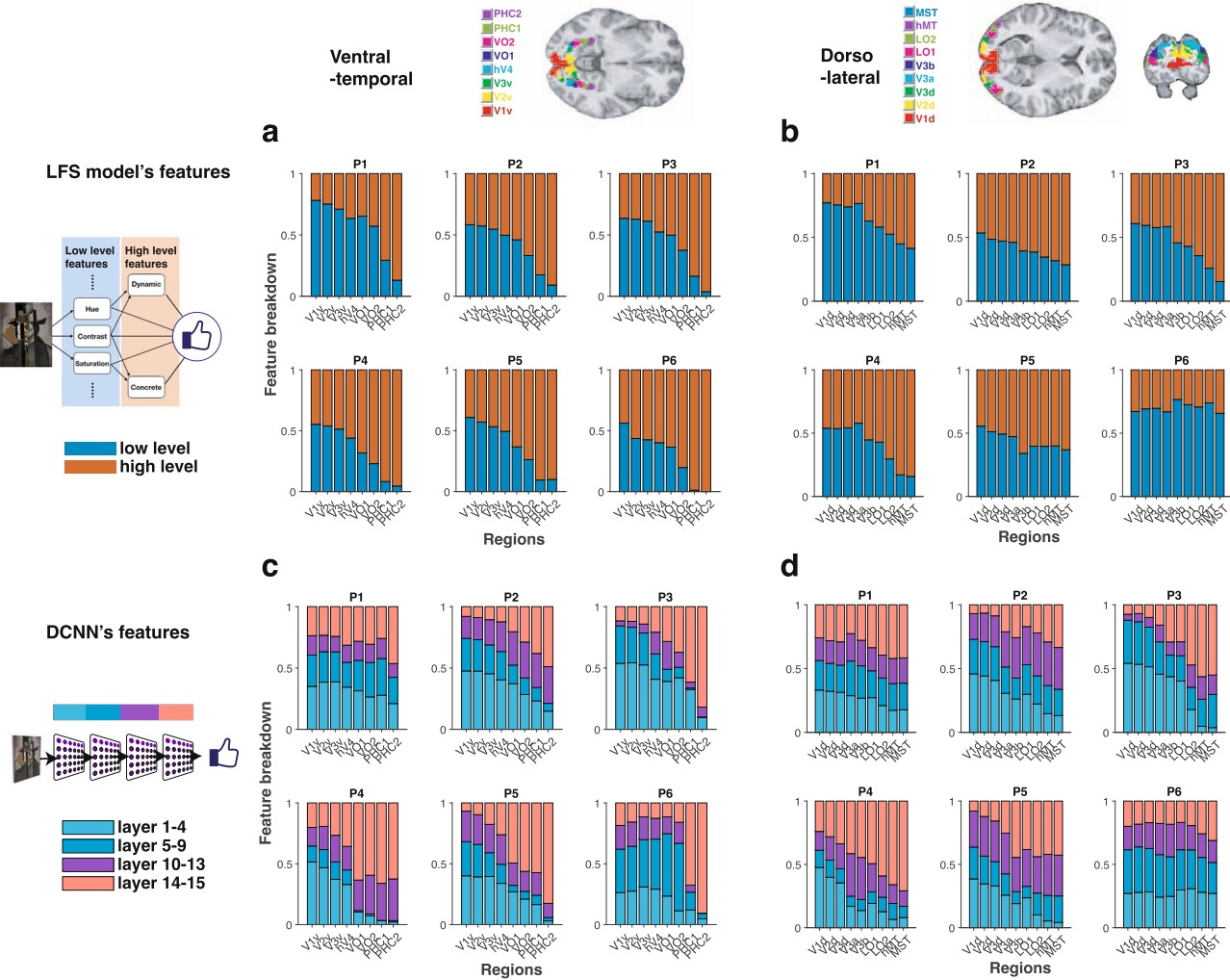

**Fig. 4 | fMRI signals in visual cortical regions show similarity to our LFS model and DCNN model. a** Encoding of low and high-level features in the visual ventral-temporal stream in a graded hierarchical manner. In general, the relative encoding of high-level features with respect to low-level features increases dramatically across the ventral-temporal stream. The maximum probabilistic map[65] is shown color-coded on the structural MR image at the top to illustrate the anatomical location of each ROI. The proportion of voxels that significantly correlated with low-level features (blue; one-sided *F*-test $p < 0.001$) against high-level features (red; one-sided *F*-test $p < 0.001$) are shown for each ROI. See the "Methods" section for detail. **b** Encoding low and high-level features in the dorsolateral visual stream. The anatomical location of each ROI[65] is color-coded on the structural MR image.

**c** Encoding of DCNN features (hidden layers' activation patterns) in the ventral-temporal stream. The top three principal components (PCs) from each layer of the DCNN were used as features in this analysis. In general, early regions more heavily encode representations found in early layers of the DCNN, while higher-order regions encode representations found in deeper CNN layers. The proportion of voxels that significantly correlated with PCs of convolutional layers 1–4 (light blue), convolutional layers 5–9 (blue), convolutional layers 10–13 (purple), fully connected layers 14–15 (pink) are shown for each ROI. The significance was set at $p < 0.001$ by one-sided *F*-test. **d** Encoding of DCNN features in the dorsolateral visual stream. Credit. Jean Metzinger, Portrait of Albert Gleizes (public domain; RISD Museum).

modestly improved the model's behavioral predictions (Supplementary Fig. 13).

## DCNN model representations
We then tested whether activity patterns in these regions resemble the computations performed by the hidden layers of the DCNN model. We extracted the first three principal components from each layer of the DCNN, and included each as regressors in a GLM. Indeed, we found evidence that both the ventral and dorsal visual streams exhibits a similar hierarchical organization to that of the DCNN, such that lower visual areas correlated better with activity in the early hidden layers of the DCNN, while higher-order visual areas (in both visual streams) tend to correlate better with activity in deeper hidden layers of the DCNN (Fig. 4c, d).

We also performed additional analyses with LASSO regression using the DCNN features. To test if we can reproduce the DCNN results

originally performed with the GLM approach (as shown in Fig. 4), we first performed LASSO regression with the same 45 features from all hidden layers. Hyperparameters were optimized by 12-fold cross-validation. The estimated weights were compared against the null distribution of each ROI constructed from the same analysis with shuffled stimuli labels. We then also performed the same analysis but with a larger set of features (150 features). In Supplementary Figs. 14 and 15, we show how the weights on features from different layers varied across different ROIs in the visual stream. We computed the sum of squared weights of hidden layer groups (layer 1–4, 5–9, 10–13, 14–15). Again, in order to discard weight estimates that can be obtained by chance, we computed a null distribution by repeating the same analysis with shuffled labels and took the weight estimates that are significantly larger than the null distribution (at $p < 0.001$) in each ROI. We again found that LASSO regression with within-subject cross validation reproduced our original GLM analysis results.

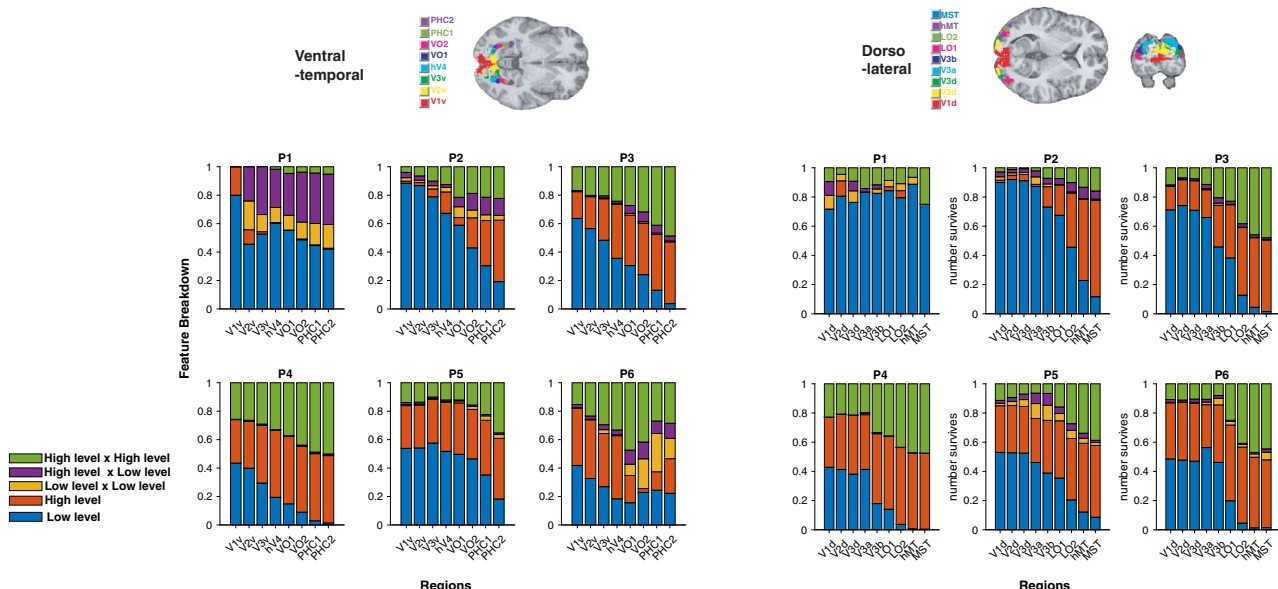

**Fig. 5 | Encoding of nonlinear feature representations.** We performed encoding analysis of low-level, high-level, and interaction term features (low × low, high × high, low × high), using lasso regression with cross validation within subject. The results of ROIs in the ventral-temporal and dorso-lateral visual streams are shown.

As a further control analysis, we asked whether similar results could be obtained from a DCNN model with random, untrained, weights[68]. We repeated the same LASSO regression analysis as we did in our analysis with the trained DCNN model. We found that such a model does not reproduce the finding of a hierarchical representation of layers that we found across the visual stream and other cortical areas as in the analysis with trained DCNN weights (Supplementary Figs. 16 and 17).

### PPC and PFC show mixed coding of low- and high-level features

We next probed these representations in downstream regions of association cortex[69,70]. We performed the same analysis with the same GLM as before in regions of interest that included the posterior parietal cortex (PPC), lateral prefrontal cortex (lPFC) and medial prefrontal cortex (mPFC). We found that both the LFS model features and the DCNN layers were represented in these regions in a mixed manner[71,72]. We found no clear evidence for a progression of the hierarchical organization that we had observed in the visual cortex; instead, each of these regions appeared to represent both low and high-level features to a similar degree (Fig. 6a). Activity in these regions also correlated with hidden layers of the DCNN model (Fig. 6b). We obtained similar results using a LASSO regression analysis with cross validation based on either the LFS model features (Supplementary Fig. 18) or the DCNN features (Supplementary Figs. 19 and 20). These findings suggest that, as we will see, these regions appear to play a primary role in feature integration as required for subjective value computations.

### Features encoded in PPC and lPFC are strongly coupled to the subjective value of visual art in mPFC

Having established that both the engineered LFS model and the emergent DCNN model features are hierarchically represented in the brain, we asked if and how these features are ultimately integrated to compute the subjective value of visual art. First, we analyzed how aesthetic value is represented across cortical regions alongside the model features by adding the participant's subjective ratings to the GLM. We found that subjective values are, in general, more strongly represented in the PPC as well as in the lateral and medial PFC than in early and late visual areas (Fig. 7a and Supplementary Fig. 21). Furthermore, value signals appeared to become more prominent in

medial prefrontal cortex compared to the lateral parietal and prefrontal regions (consistent with a large prior literature, e.g.,[11,14,17,23,31,57,73]). This pattern was not altered when we control for reaction times and the distance of individual ratings from the mean ratings, proxy measures for the degree of attention paid to each image (Supplementary Fig. 22). In a further validation of our earlier feature encoding analyses, we found that the pattern of hierarchical feature representation in visual regions was unaltered by the inclusion of ratings in the GLM (Supplementary Fig. 23). We note that even when using the DCNN model to classify features as either high or low as opposed to relying on the a-priori assignment from the LFS model, this did not change the results of our fMRI analyses qualitatively and does not affect our conclusions (Supplementary Fig. 8).

These results suggest that rich feature representations in the PPC and lateral PFC could potentially be leveraged to construct subjective values in mPFC. However, it is also possible that features represented in visual areas are directly used to construct subjective value in mPFC. To test this, we examined which of the voxels representing the LFS model features across the brain are coupled with voxels that represent subjective value in mPFC at the time when participants make decisions about the stimuli. A strong coupling would support the possibility that such feature representations are integrated at the time of decision-making in order to support a subjective value computation.

To test for this, we first performed a psychological-physiological interaction (PPI) analysis, examining which voxels are coupled with regions that represent subjective value when participants made decisions (Fig. 7b and Supplementary Fig. 24). We stress that this is not a trivial signal correlation, as in our PPI analysis all the value and feature signals are regressed out. Therefore the coupling is due to noise correlations between voxels. Then we asked how much of the feature-encoding voxels overlap with these PPI voxels. Specifically, we tested for the fraction of feature-encoding voxels that are also correlated with the PPI regressor across each ROI. Finding overlap between feature encoding voxels and PPI connectivity effects would be consistent with a role for these feature encoding representations in value construction. We found that the overlap was most prominent in the PPC and lPFC, while there was virtually no overlap in the visual areas at all (Fig. 7c), consistent with the idea that features in the PPC and lPFC, instead of visual areas, are involved in constructing subjective value

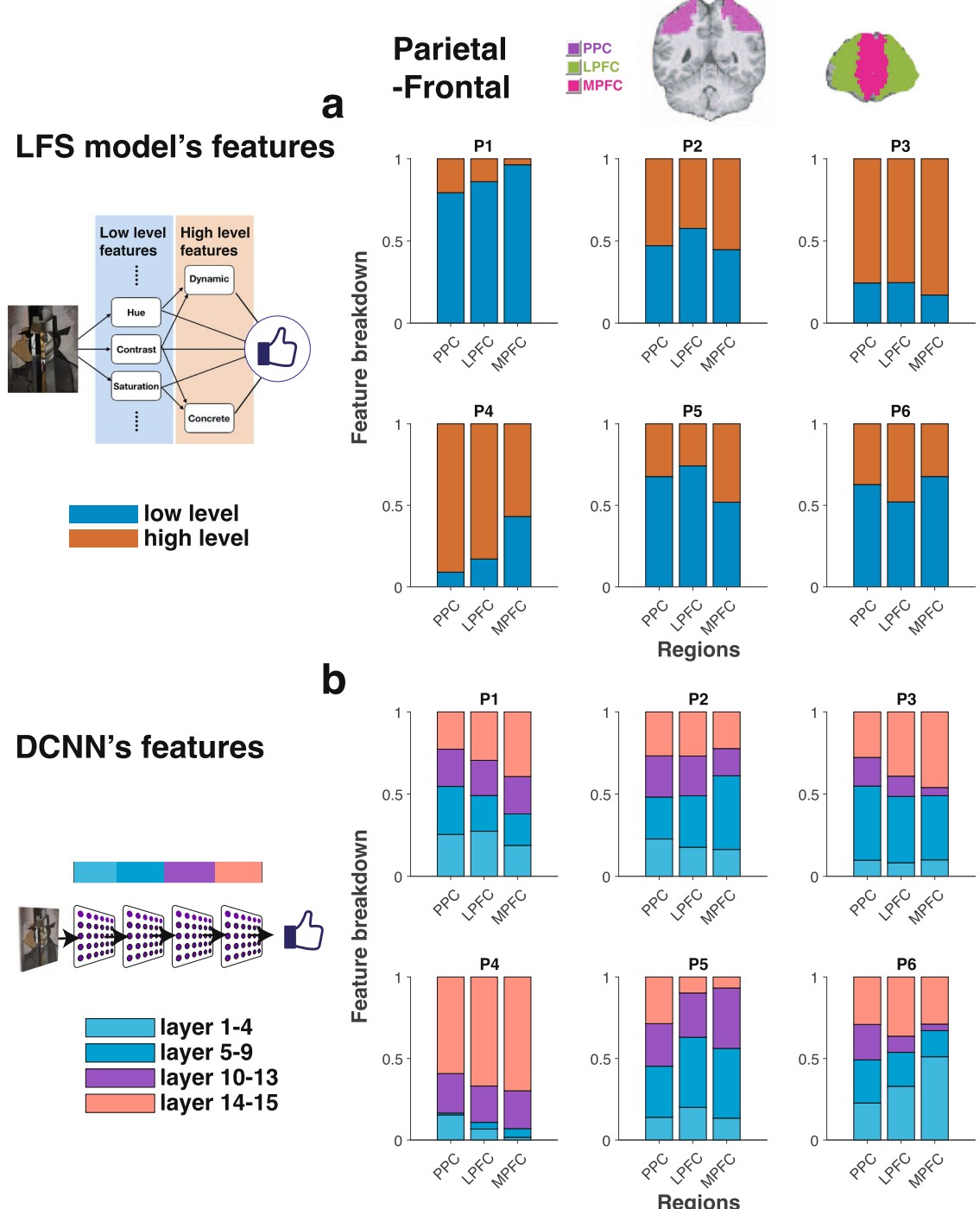

**Fig. 6 | Parietal and prefrontal cortex encode features in a mixed manner. a** Encoding of low- and high-level features from the LFS model in posterior parietal cortex (PPC), lateral prefrontal cortex (lPFC) and medial prefrontal cortex (mPFC). The ROIs used in this analysis are indicated by colors shown in a structural MR image at the top. **b** Encoding of the DCNN features (activation patterns in the hidden layers) in PPC and PFC. The same analysis method as Fig. 4 was used. Credit.Jean Metzinger, Portrait of Albert Gleizes (public domain; RISD Museum).

representations in mPFC. A more detailed decomposition of the PFC ROI from the same analysis shows the contribution of individual sub-regions of lateral and medial PFC (Supplementary Fig. 25).

We also performed a control analysis to test the specificity of the coupling to an experimental epoch by constructing a similar PPI regressor locked to the epoch of inter-trial-intervals (ITIs). This analysis showed a dramatically altered coupling that did not involve the same PPC and PFC regions (Supplementary Fig. 26). These findings indicate that coupling between PPC and LPFC with mPFC value representations

occurs specifically at the time that subjective value computations are being performed, suggesting that these regions are playing an integrative role of feature representations at the time of valuation. We however note that all of our analyses are based on correlations, which do not provide information about the direction of the coupling.

## Discussion

It is an open question how the human brain computes the value of complex stimuli such as visual art[1,3,6,74]. Here, we addressed this

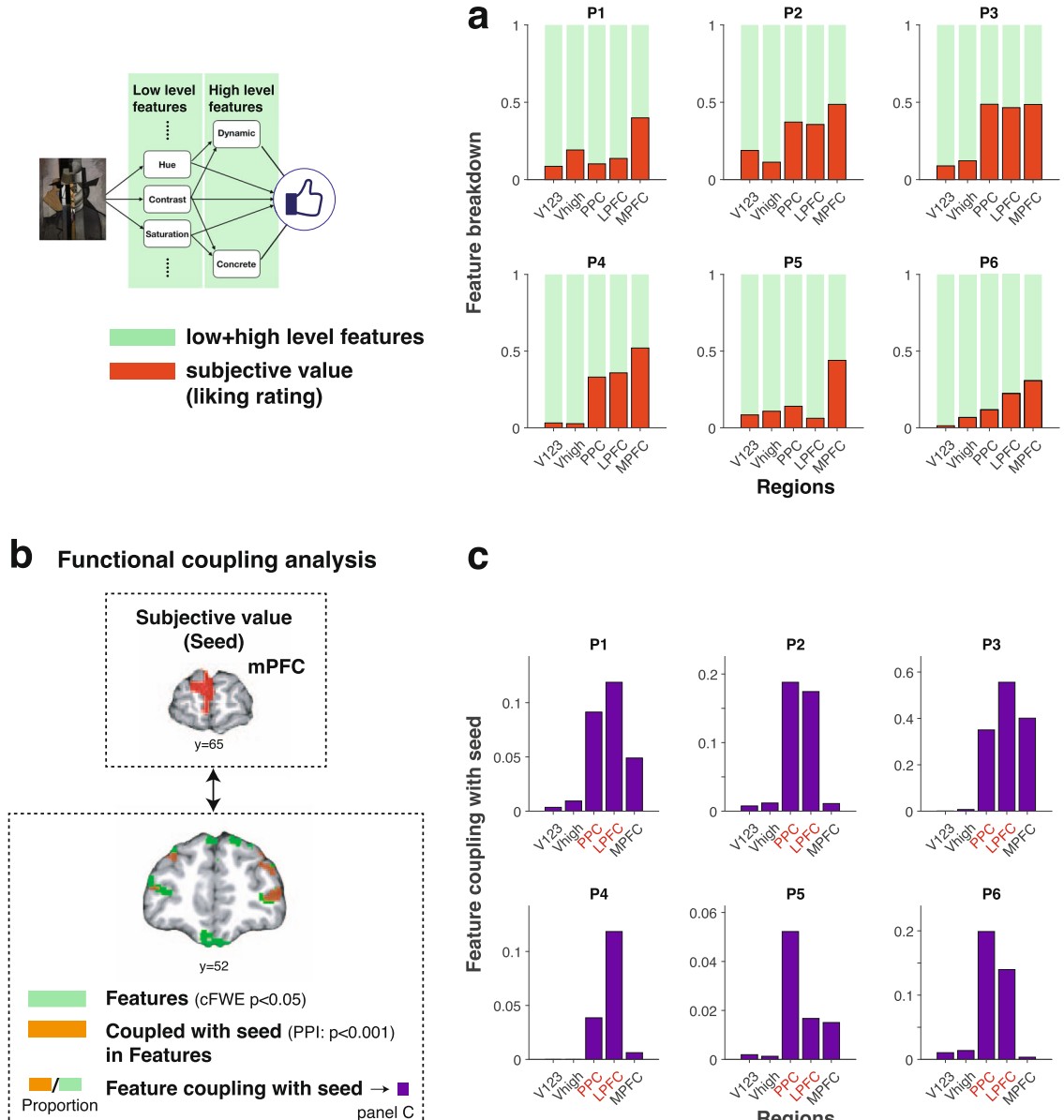

**Fig. 7 | Features are integrated from PPC and lateral PFC to medial PFC when constructing the subjective value of visual art. a** Encoding of low- and high-level features (green) and liking ratings (red) across brain regions. Note that the ROIs for the visual areas are now grouped as V1-2-3 (V1, V2 and V3) and V-high (Visual areas higher than V3). See the Methods section for detail. **b** The schematics of functional coupling analysis to test how feature representations are coupled with subjective value. We identified regions that encode features (green), by performing a one-sided *F*-test ($p < 0.05$ whole-brain cFWE with the height threshold $p < 0.001$). We also performed a psychophysiological interaction (PPI) analysis (orange: $p < 0.001$

uncorrected) to determine the regions that are coupled to the seed regions in mPFC that encode subjective value (i.e., liking rating) during stimulus presentation (red: seed, see Supplementary Fig. 24). We then tested for the proportion of overlap between voxels identified in these analyses in a given ROI. **c** The results of the functional coupling analysis show that features represented in the PPC and lPFC are coupled with the region in mPFC encoding subjective value. This result dramatically contrasts with a control analysis focusing on ITI instead of stimulus presentations (Supplementary Fig. 26). Credit. Jean Metzinger, Portrait of Albert Gleizes (public domain; RISD Museum).

question by applying two different computational models to neuroimaging data, demonstrating how the brain transforms visual stimuli into subjective value, all the way from the primary visual cortex to parietal and prefrontal cortices. The linear feature summation (LFS) model directly formulates our hypothesis, by extracting interpretable stimulus features and integrating over the features to construct subjective value. This linear regression model is related to a standard deep convolutional neural network (DCNN) trained on object recognition, because we found that the LFS model features are represented in hidden layers of the DCNN in a hierarchical manner. Here we found that both of these two models predict participants' activity across the brain from the visual cortex to prefrontal cortex. Though our

correlation-based analyses do not address the directionality of information processing across regions, our results shed light on a possible mechanism by which the brain could transform a complex visual stimulus into a simple value that informs decisions, using a rich feature space shared across stimuli.

Focusing first on the visual system, we found that low-level features that predict visual art preferences are represented more robustly in early visual cortical areas, while high-level features that predict preferences are increasingly represented in higher-order visual areas. These results support a hierarchical representation of the features required for valuation of visual imagery, and further support a model whereby lower-level features extracted by early visual regions are

integrated to produce higher-level features in the higher visual system[75]. While the notion of hierarchical representations in the visual system is well established in the domain of object recognition[76–79], our results substantially extend these findings by showing that features relevant to a very different behavioral task-forming value judgments, are also represented robustly in a similar hierarchical fashion.

We then showed that the process through which feature representations are mapped into a singular subjective value dimension in a network of brain regions, including the posterior parietal cortex (PPC), lateral and medial prefrontal cortices (lPFC and mPFC). While previous studies have hinted at the use of such a feature-based framework in the prefrontal cortex (PFC), especially in orbitofrontal cortex (OFC), in those previous studies the features were more explicit properties of a stimulus (e.g., the movement and the color of dots[45,80,81], or items that are suited to a functional decomposition such as food odor[39] or nutritive components of food[38]; see also refs. [36], [37]). Here we show that features relevant for computing subjective value of visual stimuli are widely represented in lPFC and PPC, whereas subjective value signals are more robustly represented in parietal and frontal regions, with the strongest representation in mPFC.

Further, we showed that PFC and PPC regions encoding low- and high-level features enhanced their coupling with the mPFC region encoding subjective value at the time of image presentation. While further experiments are needed to infer the directionality of the connectivity effects, our findings are compatible with a framework in which low and high-level feature representations in lPFC and PPC are utilized to construct value representations in mPFC, as we hypothesized in the LFS model.

Going beyond our original LFS model, we also found that in most participants, non-linear features created from pairs of high-level features specified in the original model produced significant correlations with neural activity across multiple regions, while largely showing similar evidence for a hierarchical organization from early to higher-order regions, as found for the linear high-level features. These findings indicate that the brain encodes a much richer set of features than our original proposed set of low-level and high-level features as specified in the original LFS model. It will be interesting to see if the nonlinear features that we introduced here, especially the ones that were constructed from pairs of high-level features, can also be used to support behavioral judgments beyond the simple value judgments studied here, such as object recognition and other more complex judgements[53]. We also note that there are other ways to construct nonlinear features. Further studies with richer set of features, e.g, other forms of interactions, may improve behavioral and neural predictions

While previous studies have suggested similarities between representations of units in DCNN models for object recognition and neural activity in the visual cortex (e.g.,[82–85]), here we show that the DCNN model can also be useful to inform how visual features are utilized for value computation across a broader expanse of the brain. Specifically, we found evidence to support the hierarchical construction of subjective value, where the early layers of DCNN correlate early areas of the visual system, and the deeper layers of DCNN correlate with higher areas of the visual system. All of the DCNN layers' information was equally represented in the PPC and PFC.

These findings are consistent with the suggestion that the hierarchical features which emerge in the visual system are projected into the PPC and PFC to form a rich feature space to construct subjective value. Further studies using neural network models with recurrent connections[46] may illuminate more detail, such as the temporal dynamics, of value construction in such a feature space across brain regions.

Accumulating evidence has suggested that value signals can be found widely across the brain including even in sensory regions (e.g.,[38,58–63]), posing a question about the differential contribution of different brain regions if value representations are so ubiquitous. While we also saw multiple brain regions that appeared to correlate with value signals during aesthetic valuation, our results suggest an alternative account for the widespread prevalence of value signals, which is that some components of the value signals especially in sensory cortex might reflect features that are ultimately used to construct value in later stages of information processing, instead of the value itself. Because neural correlates of features have not been probed previously, our results suggest that it may be possible to reinterpret at least some apparent value representations as reflecting the encoding of precursor features instead of value per se. In the present case even after taking into account feature representations, value signals were still detectable in the medial prefrontal cortex and elsewhere, supporting the notion that some brain regions are especially involved in value coding more than others. In future work it may be possible to even more clearly dissociate value from its sensory precursors by manipulating the context in which stimuli are presented, wherein features remain invariant across contexts, while the value changes. In doing so, further studies can illuminate finer dissociations between features and value signals[43].

One open question is how the brain has come to be equipped with a feature-based value construction architecture. We recently showed that a DCNN model trained solely on object recognition tasks represents the LFS's low- and high-level features in the hidden layers in a hierarchical manner, suggesting the possibility that such features could naturally emerge over development[42]. While the similarity between the DCNN and the LFS model correlations with fMRI responses in adult participants provides a promising link between these models and the brain, further investigations applying these models to studies with children or other species has the potential to inform understanding of the origin of feature-based value construction across development and across species.

Following the typical approach utilized in non-human primate and other animal neurophysiology as well as in human visual neuroimaging, we performed in-depth scanning (20 sessions) in a relatively small number of participants (six) in order to address our neural hypotheses. Because we were able to obtain a sufficient amount of fMRI data in individual participants, we were able to reliably perform single-subject inference in each participant and evaluate the results across participants side-by-side. This approach contrasts with a classic group-based neuroimaging study in which results are obtained from the group average of many participants, where each participant performs short sessions, thus providing data with low signal to noise. One advantage of our approach over the group averaging approach is that we can treat each participant as a replication unit, meaning that we can obtain multiple replications[44] from one study instead of just one group result. If every participant shows similar patterns, then it is unlikely that those results are spurious, and much more likely they reflect a true property of human brain function. We indeed found that all participants similarly performed our-hypothesized feature-based value construction across the brain. Another advantage of our methodological approach concerns possible heterogeneity across participants. Not all brains are the same, and there is known to be considerable variation in the location and morphology of different brain areas across individuals[86]. Thus, it is unlikely that all brains actually represent the same variable at the same MNI coordinates. The individual subject-based approach to fMRI analyses used here takes individual neuroanatomical variation into account, allowing for generalization that goes beyond a spatially smoothed average that does not represent any real brain. We note that one important limitation of this in-depth fMRI method is that it is not ideal for studying and characterizing differences across individuals. To gain a comprehensive account of such variability across individuals. it would be necessary to collect data from a much larger cohort of participants. As it is not feasible to scale the in-depth approach to such large cohorts due to experimenter time and

resource constraints, such individual difference studies would necessarily require adopting more standard group-level scanning approaches and analyses.

While we found that results from the visual cortex were largely consistent across participants, the proportion of features represented in PCC and PFC, as well as the features that were used, were quite different across participants. Understanding such individual differences will be important in future work. For instance, there is evidence that art experts tend to evaluate art differently from people with no artistic training[87,88]. It would be interesting to study if feature representations may differ between experts and non-experts, while probing whether the computational motif that we found here (hierarchical visual feature representation in visual areas, value construction in PPC and PFC) might be conserved across different levels of expertise. We should also note that the model's predictive accuracy about liking ratings varied across participants. It is likely that some participants used features that our model did not consider, such as personal experience associated with stimuli. Brain regions such as the hippocampus may potentially be involved in such additional feature computations. Further, behavior and fMRI signals can be inherently noisy in that there will be a portion of data that cannot be predicted (i.e. a noise ceiling). Characterizing the contribution of these noise components will require further experiments with repeated measurements of decisions about the same stimuli.

Taken together, these findings are consistent with the existence of a large-scale processing hierarchy in the brain that extends from early visual cortex to medial prefrontal cortex, whereby visual inputs are transformed into various features through the visual stream. These features are then projected to PPC and lPFC, and subsequently integrated into subjective value judgment in mPFC. Crucially, the flexibility afforded by such a feature-based mechanism of value construction ensures that value judgments can be formed even for stimuli that have never before been seen, or in circumstances where the goal of valuation varies (e.g., selecting a piece of art as a gift). Therefore, our study proposes a brain-wide computational mechanism that does not limit to aesthetics, but can be generalized to value constrictions of a wide range of visual and other sensory stimuli.

## Methods

### Participants
All participants provided informed consent for their participation in the study, which was approved by the Caltech IRB.

Six volunteers (female: 6; age 18–24 yr: 4; 25–34 yr: 1; 35–44 yr: 1. 4 White, 2 Asian) were recruited into our fMRI study. one participant completed master's degree or higher, four participants earned a college degree as the highest level, and one participant had a high-school degree as the highest degree. None of the participants possessed an art degree. All of the participants reported that they visit art museums less than once a month.

In addition, thirteen art-experienced participants [reported in our previous behavioral paper[42]] (female: 6; ages 18–24 yr: 3; 25–34 yr: 9; 35–44 yr: 1) were invited to evaluate the high-level feature values (outside the scanner). These participants for annotation were primarily recruited from the ArtCenter College of Design community.

### Stimuli
The same stimuli as our recent behavioral study[42] were used in the current fMRI study. Painting stimuli were taken from the visual art encyclopedia www.wikiart.org. Using a script that randomly selects images in a given category of art, we downloaded 206 or 207 images from four categories of art (825 in total). The categories were 'Abstract Art', 'Impressionism', 'Color Fields', and 'Cubism'. We randomly downloaded images with each tag using our custom code in order to avoid subjective bias. We supplemented this database with an additional 176 paintings that were used in a previous study[52]. For the fMRI study reported here, one image was excluded from the full set of 1001 images to have an equal number of trials per run (50 images/run × 20 runs = 1000 images).

### fMRI task
On each trial, participants were presented with an image of the artwork on the computer screen for three seconds. Participants were then presented with a scale from 0, 1, 2, 3 in which they had to indicate how much they liked the artwork. The location of each numerical score was randomized across trials. Participants had to press a button of a button box that they hold with both hands to indicate their rating within three seconds, where each of four buttons corresponded to a particular location on the screen from left to right. The left (right) two buttons were instructed to be pressed by their left (right) thumb. After a brief feedback period showing their chosen rating (0.5 s), a center cross was shown for inter-trial intervals (jittered between 2 and 9 s). Each run consists of 50 trials. Participants were invited to the study over four days to complete twenty runs, where participants completed on average five runs on each day.

### fMRI data acquisition
fMRI data were acquired on a Siemens Prisma 3T scanner at the Caltech Brain Imaging Center (Pasadena, CA). With a 32-channel radio-frequency coil, a multi-band echo-planar imaging (EPI) sequence was employed with the following parameters: 72 axial slices (whole-brain), A-P phase encoding, −30 degrees slice tilt with respect to AC-PC line, echo time (TE) of 30ms, multi-band acceleration of 4, repetition time (TR) of 1.12 s, 54-degree flip angle, 2 mm isotropic resolution, echo spacing of 0.56 ms. 192 mm × 192 mm field of view, in-plane acceleration factor 2, multi-band slice acceleration factor 4.

Positive and negative polarity EPI-based field maps were collected before each run with very similar factors as the functional sequence described above (same acquisition box, number of slices, resolution, echo spacing, bandwidth and EPI factor), single band, TE of 50 ms, TR of 5.13 s, 90-degree flip angle.

T1-weighted and T2-weighted structural images were also acquired once for each participant with 0.9 mm isotropic resolution. T1's parameters were: repetition time (TR) 2.4 s: echo time (TE), 0.00232 s; inversion time (TI) 0.8 s; flip angle, 10 degrees; in-plane acceleration factor 2. T2's parameters were: TR 3.2 s; TE 0.564 s; flip angle, 120 degrees; in-plane acceleration factor 2.

### fMRI data processing
Results included in this manuscript come from preprocessing performed using *fMRIPrep* 1.3.2 (ref. 89; RRID:SCR_016216), which is based on *Nipype* 1.1.9 (ref. 90; RRID:SCR_002502).

**Anatomical data preprocessing.** The T1-weighted (T1w) image was corrected for intensity non-uniformity (INU) with N4Bias Field Correction[91], distributed with ANTs 2.2.0 [[92], RRID:SCR_004757] and used as T1w-reference throughout the workflow. The T1w-reference was then skull-stripped with a *Nipype* implementation of the antsBrainExtraction.sh workflow (from ANTs), using OASIS30ANTs as target template. Spatial normalization to the *ICBM 152 Nonlinear Asymmetrical template version 2009c* was performed through nonlinear registration with antsRegistration (ANTs 2.2.0), using brain-extracted versions of both T1w volume and template. Brain tissue segmentation of cerebrospinal fluid (CSF), white-matter (WM) and gray-matter (GM) was performed on the brain-extracted T1w using fast.

**Functional data preprocessing.** For each of the 20 BOLD runs found per subject (across all tasks and sessions), the following preprocessing was performed. First, a reference volume and its skull-stripped version were generated using a custom methodology of *fMRIPrep*. A

deformation field to correct for susceptibility distortions was estimated based on two echo-planar imaging (EPI) references with opposing phase-encoding directions, using `3dQwarp` (AFNI 20160207). Based on the estimated susceptibility distortion, an unwarped BOLD reference was calculated for a more accurate co-registration with the anatomical reference.

The BOLD reference was then co-registered to the T1w reference using `flirt` with the boundary-based registration cost-function. Co-registration was configured with nine degrees of freedom to account for distortions remaining in the BOLD reference. Head-motion parameters with respect to the BOLD reference (transformation matrices, and six corresponding rotation and translation parameters) are estimated before any spatiotemporal filtering using `mcflirt` The BOLD time-series (including slice-timing correction when applied) were resampled onto their original, native space by applying a single, composite transform to correct for head-motion and susceptibility distortions. These resampled BOLD time series will be referred to as preprocessed BOLD in original space, or just preprocessed BOLD. The BOLD time series were resampled to MNI152NLin2009cAsym standard space, generating a preprocessed BOLD run in MNI152NLin2009cAsym space. First, a reference volume and its skull-stripped version were generated using a custom methodology of *fMRIPrep*. Several confounding time series were calculated based on the preprocessed BOLD: framewise displacement (FD), DVARS and three region-wise global signals. FD and DVARS are calculated for each functional run, both using their implementations in *Nipype*.

The three global signals are extracted within the CSF, the WM, and the whole-brain masks. In addition, a set of physiological regressors were extracted to allow for component-based noise correction. Principal components are estimated after high-pass filtering the *preprocessed BOLD* time series (using a discrete cosine filter with 128 s cut-off) for the two *CompCor* variants: temporal (tCompCor) and anatomical (aCompCor). Six tCompCor components are then calculated from the top 5% variable voxels within a mask covering the subcortical regions. This subcortical mask is obtained by heavily eroding the brain mask, which ensures it does not include cortical GM regions. For aCompCor, six components are calculated within the intersection of the aforementioned mask and the union of CSF and WM masks calculated in T1w space, after their projection to the native space of each functional run (using the inverse BOLD-to-T1w transformation).

The head-motion estimates calculated in the correction step were also placed within the corresponding confounds file. All resamplings can be performed with a single interpolation step by composing all the pertinent transformations (i.e. head-motion transform matrices, susceptibility distortion correction when available, and co-registrations to anatomical and template spaces). Gridded (volumetric) resamplings were performed using ants Apply Transforms (ANTs), configured with Lanczos interpolation to minimize the smoothing effects of other kernels.

### Computational models
The computational methods and behavioral modeling reported in this manuscript overlap with that reported in our recent article focusing exclusively on behavior[42]. For completeness, we reproduce some of the descriptions of these methods as first described in ref. 42.

### Linear feature summation model (LFS model)
We hypothesized that subjective preferences for visual stimuli are constructed by the influence of visual and emotional features of the stimuli. As its simplest, we assumed that the subjective value of the $i$-th stimulus $v_i$ is computed by a weighted sum of feature values $f_{i,j}$:

$$v_i = \sum_{j=0}^{n_f} w_j f_{i,j} \qquad (1)$$

where $w_j$ is a weight of the $j$-th feature, $f_{i,j}$ is the value of the $j$-th feature for stimulus $i$, and $n_f$ is the number of features. The 0-th feature is a constant $f_{i,0} = 1$ for all $i$'s.

Importantly, $w_j$ is not a function of a particular stimulus but shared across all visual stimuli, reflecting the *taste* of a participant. The same taste ($w_j$'s) can also be shared across different participants, as we showed in our behavioral analysis. The features $f_{i,j}$ were computed using visual stimuli; we used the same feature values to predict liking ratings across participants. We used the simple linear model Eq. (1) to predict liking ratings in our behavioral analysis (see below for how we determined features and weights).

As we schematically showed in Fig. 1, we hypothesized that the input stimulus is first broke down into low-level features and then transformed into high-level features, and indeed we found that a significant variance of high-level features can be predicted by a set of low-level features. This hierarchical structure of the LFS model was further tested in our DCNN and fMRI analysis.

### Features
Because we did not know a priori what features would best describe human aesthetic values for visual art, we constructed a large feature set using previously published methods from computer vision augmented with additional features that we ourselves identified using additional existing machine learning methods.

**Visual low-level features introduced in ref. 49.** We employed 40 visual features introduced in ref. 49. We do not repeat descriptions of the features here; but briefly, the feature sets consist of 12 global features that are computed from the entire image that include color distributions, brightness effects, blurring effects, and edge detection, and 28 local features that are computed for separate segments of the image (the first, the second and the third largest segments). Most features are computed straightforwardly in either HSL (hue, saturation, lightness) or HSV (hue, saturation, value) space (e.g., average hue value).

One feature that deserves description is a blurring effect. Following[49,93], we assumed that the image $I$ was generated from a hypothetical sharp image with a Gaussian smoothing filter with an unknown variance $\sigma$. Assuming that the frequency distribution for the hypothetical image is approximately the same as the blurred, actual image, the parameter $\sigma$ represents the degree to which the image was blurred. The $\sigma$ was estimated by the Fourier transform of the original image by the highest frequency, whose power is greater than a certain threshold.

$$f_{\text{blur}} = max\left(k_x, k_y\right) \propto \frac{1}{\sigma} \qquad (2)$$

where $k_x = 2(x - n_x/2)/n_x$ and $k_y = 2(y - n_y/2)/n_y$ with $(x, y)$ and $(n_x, n_y)$ are the coordinates of the pixel and the total number of pixel values, respectively. The above max was taken within the components whose power is larger than four[49].

The segmentation for this feature set was computed by a technique called kernel GraphCut[50,94]. Following ref. 49, we generated a total of at least six segments for each image using a $C^{++}$ and Matlab package for kernel graph cut segmentation[94]. The regularization parameter that weighs the cost of cut against smoothness was adjusted for each image in order to obtain about six segments. See refs. 49, 94 for the full description of this method and examples.

Of these 40 features, we included all of them in our initial feature set except for local features for the third-largest segment, which were highly correlated with features for the first and second-largest segments and were thus deemed unlikely to add unique variance to the feature prediction stage.

**Additional low-level features.** We assembled the following low-level features to supplement the set by Li & Chen[49]. These include both global features and local features. Local features were calculated on segments determined by two methods. The first method was statistical region merging (SRM) as implemented by ref. 95, where the segmentation parameter was incremented until at least three segments were calculated. The second method converted paintings into LAB color space and used k-means clustering of the A and B components. While the first method reliably identified distinct shapes in the paintings, the second method reliably identified distinct color motifs in the paintings.

The segmentation method for each feature is indicated in the following descriptions. Each local feature was calculated on the first and second-largest segments.

Local features:

- Segment size (SRM): Segment size for segment $i$ was calculated as the area of segment i over the area of the entire image:

$$f_{\text{segment size}} = \frac{\text{area segment } i}{\text{total area}} \quad (3)$$

- HSV mean (SRM): To calculate mean hue, saturation and color value for each segment, segments were converted from RGB to HSV color space.

$$f_{\text{mean hue}} = \text{mean(hue values in segment } i) \quad (4)$$

$$f_{\text{mean saturation}} = \text{mean(saturation values in segment } i) \quad (5)$$

$$f_{\text{mean colorvalue}} = \text{mean(color values in segment } i) \quad (6)$$

- Segment moments (SRM):

$$f_{\text{CoMX coordinate}} = \frac{\sum_{k \in \text{segment } i} x_k}{\text{area segment } i} \quad (7)$$

$$f_{\text{CoMY coordinate}} = \frac{\sum_{k \in \text{segment } i} y_k}{\text{area segment } i} \quad (8)$$

$$f_{\text{Variance}} = \frac{\sum_{k \in \text{segment } i} (x_k - \bar{x})^2 + (y_k - \bar{y})^2}{\text{area segment } i} \quad (9)$$

$$f_{\text{Skew}} = \frac{\sum_{k \in \text{segment } i} (x_k - \bar{x})^3 + (y_k - \bar{y})^3}{\text{area segment } i} \quad (10)$$

where $(\bar{x}, \bar{y})$ is the center of mass coordinates of the corresponding segment.

- Entropy (SRM):

$$f_{\text{entropy}} = -\sum_j (p_j * \log_2(p_j)) \quad (11)$$

where $p$ equals the normalized intensity histogram counts of segment $i$.

- Symmetry (SRM): For each segment, the painting was cropped to maximum dimensions of the segment. The horizontal and vertical mirror images of the rectangle were taken, and the mean squared error of each was calculated from the original.

$$f_{\text{horizontal symmetry}} = \frac{\sum_{x,y \in \text{segment}} (\text{segment}_{x,y} - \text{horizontal\_flip(segment)}_{x,y})^2}{\text{\# pixels in segment}} \quad (12)$$

$$f_{\text{vertical symmetry}} = \frac{\sum_{x,y \in \text{segment}} (\text{segment}_{x,y} - \text{vertical\_flip(segment)}_{x,y})^2}{\text{\# pixels in segment}} \quad (13)$$

- R-value mean (K-means): Originally, we took the mean of R, G, and B values for each segment, but found these values to be highly correlated, so we reduced these three features down to just one feature for mean R value.

$$f_{\text{R-value}} = \text{mean(R} - \text{values in segment)} \quad (14)$$

- HSV mean (K-means): As with SRM-generated segments, we took the hue, saturation, and color value means of segments generated by K-means segmentation as described in equations 2–4.

Global features:

- Image intensity: Paintings were converted from RGB to grayscale from 0 to 255 to yield a measure of intensity. The 0-255 scale was divided into five equally sized bins. Each bin count accounted for one feature.

$$f_{\text{intensity count bin } i \in \{1,4\}} = \frac{\text{\# pixels with intensity} \in \left\{ \frac{255(i-1)}{5}, \frac{255i}{5} \right\}}{\text{total area}} \quad (15)$$

- HSV modes: Paintings were converted to HSV space, and the modes of the hue, saturation, and color value across the entire painting were calculated. While we took mean HSV values over segments in an effort to calculate overall-segment statistics, we took the mode HSV values across the entire image in an effort to extract dominating trends across the painting as a whole.

$$f_{\text{modehue}} = \text{mode(hue values in segment } i) \quad (16)$$

$$f_{\text{modesaturation}} = \text{mode(saturation values in segment } i) \quad (17)$$

$$f_{\text{modecolorvalue}} = \text{mode(color values in segment } i) \quad (18)$$

- Aspect (width-height) Ratio:

$$f_{\text{aspectratio}} = \frac{\text{image width}}{\text{image height}} \quad (19)$$

- Entropy: Entropy over the entire painting was calculated according to Eq. (9).

**High-level feature set**[51,52]. We also introduced features that are more abstract and not easily computed by a simple algorithm. In ref. 51, Chatterjee et al. pioneered this by introducing 12 features (color temperature, depth, abstract, realism, balance, accuracy, stroke, animacy, emotion, color saturation, complexity) that were annotated by human participants for 24 paintings, in which the authors have found that annotations were consistent across participants, regardless of their artistic experience. Vaidya et al.[52] further collected annotations of these feature sets from artistically experienced participants for an additional 175 paintings and performed a principal component analysis, finding three major components that summarize the variance of the original 12 features. Inspired by the three principal components, we introduced three high-level features: concreteness, dynamics, and temperature. Also, we introduced valence as an additional high-level feature. The four high-level features were annotated in a similar manner to the previous studies[51,52]. We took the mean annotations of all 13 participants for each image as feature values. In addition, we also annotated our image set with whether or not each image included a person. This was done by manual annotation, but it can also be done with a human detection algorithm (e.g., see ref. 96). We included this presence-of-a-person feature in the low-level feature set originally[97], though we found in our DCNN analysis that the feature shows a signature of a high-level feature[97]. Therefore in this current study, we included this presence of a person to the high-level feature set. As we showed in the main text, classifying this feature as a low-level feature or as a high-level feature does not change our results.

### Identifying the shared feature set that predicts aesthetic preferences

The above method allowed us to have a set of 83 features in total that are possibly used to predict human aesthetic valuation. These features are likely redundant because some of them are highly correlated, and many may not contribute to decisions at all. We thus sought to identify a minimal subset of features that are commonly used by participants. In ref. 97, we performed this analysis using Matlab Sparse Gradient Descent Library (https://github.com/hiroyuki-kasai/SparseGDLibrary). For this, we first orthogonalized features by sparse PCA[98]. Then we performed a regression with a LASSO penalty at the group level using participants' behavioral data with a function $group - lasso - problem$. We used Fast Iterative Soft Thresholding Algorithm (FISTA) with cross-validation. After eliminating PC's that were not shared by more than one participant, we transformed the PC's back to the original space. We then eliminated one of the two features that were most highly correlated ($r^2 > 0.5$) to obtain the final set of shared features.

To identify relevant features for use in the current fMRI analysis, we utilized behavioral data from both our previous in-lab behavioral study (ref. 97 and the fMRI participants included in the current study (13 participants in total). Because the goal of the fMRI analysis is to highlight the hierarchical nature in neural coding between low and high-level features, we first repeated the above procedure with low-level features alone (79 features in total) and then we added high-level features (the concreteness, the dynamics, the temperature, and the valance) to the obtained shared low-level features.

The identified shared features are the following: the concreteness, the dynamics, the temperature, the valence, the global average saturation from ref. 49, the global blurring effect from ref. 49, the horizontal coordinate of mass center for the largest segment using the Graph-cut from ref. 49, the vertical coordinate of mass center for the largest segment using the Graph-cut from ref. 49, the mass skewness for the second largest segment using the Graph-cut from ref. 49, the size of the largest segment using SRM, the mean hue of the largest segment using SRM, the mean color value of largest segment using SRM, the mass variance of the largest segment using SRM, global entropy, the entropy of the second-largest segment using SRM, the image intensity in bin 1, the image intensity in bin 2, and the presence of a person.

**Nonlinear interaction features**. We constructed additional feature sets by multiplying pairs of LFS features. We grouped the resulting features into three groups. (1) features created from interactions between high-level features (2) features created from interactions between low-level features and (3) features created from interactions between a high-level and a low-level feature. In order to determine the contribution of these three groups of features, we performed PCA on each group so that we can take the same number of components from each group. In our analysis, we took five PCs from each group to match with the number of features of original high-level features.

### Behavioral model fitting

We tested how our shared-feature model can predict human liking ratings using out-of-sample tests. All models were cross-validated in twenty folds, and we used ridge regression unless otherwise stated. Hyperparameters were tuned by cross-validation. We calculated the Pearson correlation between model predictions (pooled predictions from all cross-validation sets) and actual data, and defined it as the predictive accuracy.

We estimated individual participant's feature weights by fitting a linear regression model with the shared feature set to each participant. For illustrative purposes, the weights were normalized for each participant by the maximum feature value (concreteness) in Fig. 1g, Supplementary Figs. 1, 12.

The significance of the above analyses was measured by generating a null distribution constructed by the same analyses but with permuted image labels. The null distribution was construed by 10,000 permutations. The chance level was determined by the mean of the null distribution.

### Deep convolutional neural network (DCNN) analysis

**Network architecture**. The deep convolutional neural network (DCNN) we used consists of two parts. An input image feeds into convolutional layers from the standard VGG-16 network that is pre-trained on ImageNet. The output of the convolutional layers then projects to fully connected layers. This architecture follows the current state-of-the-art model on aesthetic evaluation[99,100].

The details of the convolutional layers from the VGG network can be found in ref. 54; but briefly, it consists of 13 convolutional layers and 5 intervening max pooling layers. Each convolutional layer is followed by a rectified linear unit (ReLU). The output of the final convolutional layer is flattened to a 25088-dimensional vector so that it can be fed into the fully connected layer.

The fully connected part has two hidden layers, where each layer has 4096 dimensions. The fully connected layers are also followed by a ReLU layer. During training, a dropout layer was added with a drop out probability 0.5 after every ReLU layer for regularization. Following the current state of the art model[100], the output of the fully connected network is a 10-dimensional vector that is normalized by a softmax. The output vector was weighted averaged to produce a scalar value[100] that ranges from 0 to 3.

**Network training**. We trained our model on our behavioral data set by tuning weights in the fully connected layers. We employed 10-fold cross-validation to benchmark the art rating prediction. The model was optimized using a Huber loss metric, which is robust to outliers[101]. We used stochastic gradient descent (SGD) with momentum to train the model. We used a batch size of 100, a learning rate of $10^{-4}$, the momentum of 0.9, and weight decay of $5 \times 10^{-4}$. The learning rate decayed by a factor of 0.1 every 30 epochs.

To handle various sizes of images, we used the zero-padding method. Because our model could only have a $224 \times 224$ sized input, we first scaled the input images to have the longer edges be 224 pixels long. Then we filled the remaining space with 0 valued pixels (black).

We used Python 3.7, Pytorch 0.4.1.post2, and CUDA 9.0 throughout the analysis.

**Retraining DCNN to extract hidden layer activations.** We also trained our network on single-fold ART data in order to obtain a single set of hidden layer activations. We prevented over-fitting by stopping our training when the model performance (Pearson correlation between the model's prediction and data) reached the mean correlation from the 10-folds cross-validation.

**Decoding features from the deep neural network.** We decoded the LFS model features from hidden layers by using linear (for continuous features) and logistic (for categorical features) regression models, as we described in ref. 97. We considered the activations of outputs of ReLU layers (total of 15 layers). First, we split the data into ten folds for the 10-fold cross-validation. In each iteration of the cross-validation, because dimensions of the hidden layers are much larger $(64 \times 224 \times 224 = 3,211,264)$ than the actual data size, we first performed PCA on the activation of each hidden layer from the training set. The number of principal components was chosen to account for 80% of the total variance. By doing so, each layer's dimension was reduced to less than 536. Then the hidden layers' activations from the test set were projected onto the principal component space by using the fitted PCA transformation matrices. The hyperparameter of the ridge regression was tuned by doing a grid search, and the best-performing coefficient for each layer and feature was chosen based on the scores from the 10-folds cross-validation. We tested for a total of 19 features, including all 18 features that we used for our fMRI analysis, as well as the simplest feature that was not included into our fMRI analysis (as a result of our group-level feature selection) but that was also of interest here: the average hue value. For the continuous features (e.g., rating, mean hue), Pearson correlation between the model's prediction and data were used as the metric for the goodness of fit, while for the categorical features (e.g., presence of person), we calculated accuracy, the area under curve (AUC), and F1 scores. The sign of slopes of decoding plots from these metrics were identical. In a supplementary analysis, we also explored whether adding 'style matrices' of hidden layers[102] to the PCA-transformed hidden layer's activations can improve the decoding accuracy; however, we found the style matrices do not improve the decoding accuracy. Sklearn 0.19.2 on Python 3.7 was used for the decoding analyses.

**Reclassifying features according to the slopes of the decoding accuracy across hidden layers.** In our LFS model, we classified putative low-level and high-level features simply by whether a feature is computed by a computer algorithm vs annotated by humans respectively. In reality, however, some putative low-level features are more complex in terms of how they could be constructed than other lower-level features, while some putative high-level features could in fact be computed straightforwardly from raw pixel inputs. Using the decoding results of the features from hidden layers in the DCNN, we identified DCNN-defined low-level and high-level features. For this, we fit a linear slope to the estimated decoding accuracy vs hidden layers. We permuted layer labels 10,000 times and performed the same analysis to construct null distribution as described earlier. We classified a feature as high-level if the slope was significantly positive at $p < 0.001$, and we classified a feature as a low-level feature if the slope was significantly negative at $p < 0.001$.

The features showing negative slopes were: the average hue, the average saturation, the average hue of the largest segment using GraphCut, the average color value of the largest segment using GraphCut, the image intensity in bin 1, the image intensity in bin 3, and the temperature.

The features showing positive slopes were: the concreteness, the dynamics, the presence of a person, the vertical coordinate of the mass center for the largest segment using the Graph Cut, the mass variance of the largest segment using the SRM, the entropy in the 2nd largest segment using SRM. All of these require relatively complex computations, such as localization of segments or image identification. This is consistent with a previous study showing that object-related local features showed a similar increased decodability at a deeper layer[85].

## fMRI analysis

**Standard GLM analysis.** We conducted a standard GLM analysis on the fMRI data with SPM 12. The SPM feature for asymmetrically orthogonalizing parametric regressors was disabled throughout. We collected enough data from each individual participant (four days of scanning) so that we can analyze and interpret each participant's results separately. The following regressors were obtained from the fmriprep preprocessing pipeline and added to all analysis as nuisance regressors: framewise displacement, comp-cor, non-steady, trans, rot. The onsets of stimulus, decision, and action were also controlled by stick regressors in all GLMs described below. In addition, we added the onset of the Decision period, the onset of feedback to all GLM as nuisance regressors, because we focused on the stimulus presentation period.

**Identifying subjective value coding (GLM 1).** In order to gain insight into how the subjective value of art was represented in the brain, we performed a simple GLM analysis with a parametric regressor at the onset of the Stimulus (GLM 1). The parameter was linearly modulated by participant's liking ratings on each trial. The results were cluster FWE collected with a height threshold of $p < 0.001$.

**Identifying feature coding (GLM 2, 3, 2', 3').** In order to gain insight into how features were represented in the brain, we performed another GLM analysis with a parametric regressor at the onset of Stimulus (GLM 2, 3). In GLM 2, there are in total 18 feature-modulated regressors; each representing the value of one of the shared features for the fMRI analysis. We then performed F-tests on high-level features and low-level features (a diagonal contrast matrix with an entry set to one for each feature of interest was constructed in SPM) in order to test whether a voxel is significantly modulated by any of the high and/or low-level features. We then counted the number of voxels that are significantly correlated $(p < 0.001)$ in each ROI (note that the F-value for significance is different for high and low features due to the difference in the number of consisting features). We then displayed the proportions of two numbers in a given ROI.

We performed a similar analysis using the DCNN hidden layers (GLM 3). We took the first three principal components of each convolutional and fully connected layers (three PCs times 15 layers = 45 parametric regressors). We then performed F-tests on PCs from layers 1 to 4, layers 5 to 9, layers 10 to 13, and fully connected layers (layers 14 and 15). The proportions of the survived voxels were computed for each ROI.

In addition, we also performed the same analyses with GLMs to which we added liking ratings for each stimulus. We call these analyses GLM 2' and GLM 3', respectively.

We note that, because in our LFS model the liking rating is a linear integration of features, adding liking rating regressor means to identify neural correlates of the liking ratings that are outside of the LFS model's prediction.

**Region of interests (ROI).** We constructed ROIs for visual topographic areas using a previously published probabilistic map[65]. We constructed 17 masks based on the 17 probabilistic maps taken from ref. 65, consisting of 8 ventral-temporal (V1v, V2v, V3v, hV4, VO1, VO2, PHC1, and PHC2) and 9 dorsal-lateral (V1d, V2d, V3d, V3A, V3B, LO1, LO2, hMT, and MST) masks. In this, ventral and dorsal regions for early visual areas V1, V2, V3 are separately defined. Each mask was constructed by

thresholding the probability map at $p > 0.01$. We defined $V_{123}$ as V1v +V2v+ V3v+ V1d + V2d + V3d + V3A + V3B, and $V_{high}$ as hV4 + VO1 + VO2 + PHC1 + PHC2 + LO1 + LO2 + hMT + MST. (hV4: human V4, VO: ventral occipital cortex, PHC: posterior parahippocampal cortex, LO: lateral occipital cortex, hMT: human middle temporal area, MST: medial superior temporal area.)

We also constructed ROIs for parietal and prefrontal cortices using the AAL database. Posterior parietal cortex (PPC) was defined by bilateral MNI-Parietal-Inf + MNI-Parietal-Sup. lateral orbitofrontal cortex (lOFC) was defined by bilateral MNI-Frontal-Mid-Orb + MNI-Frontal-Inf-Orb + MNI-Frontal-Sup-Orb, and medial OFC (mOFC) was defined by bilateral MNI-Frontal-Med-Orb + bilateral MNI-Rectus. Dorsomedial PFC (dmPFC) was defined by bilateral MNI-Frontal-Sup-Medial + MNI-Cingulum-Ant, and dorsolateral PFC (dlPFC) was defined by bilateral MNI-Frontal-Mid + MNI-Frontal-Sup. Ventrolateral PFC (vlPFC) was defined by bilateral MNI-Frontal-Inf-Oper + MNI-Frontal-Inf-Tri.

We also constructed lateral PFC (LPFC) as vlPFC + dlPFC +lOFC, and medial PFC (MPFC) as mOFC + dmPFC.

**PPI analysis (GLM 4, 4′).** We conducted a psychobiological-physiological interaction analysis. We took a seed from the GLM 1 identified cluster showing subjective value in MPFC (Supplementary Fig. 24), and a psychological regressor as a box function, which is set to one during the stimulus epoch and 0 otherwise. We added the time course of the seed, the PPI regressor, to a variant of GLM 2′ (the parametric regressors in which feature values and liking values were constructed using a boxcar function at stimulus periods, instead of its onsets) and determined which voxels were correlated with the PPI regressor (GLM 4). Following ref. 38, boxcar functions were used because feature integration can take place throughout the duration of each stimulus presentation.

We also conducted a control PPI analysis. For this we took the same seed, but now the psychological regressor was a box function which is one during ITI, and 0 otherwise. We added the time course of the seed and the PPI regressor, the box function for ITI, and the PPI regressor to the same variant of GLM 2′ (the parametric regressors with feature and liking values were constructed using boxcar function at Stimulus periods, instead of its onsets). We refer to this as GLM 4′.

**Feature integration analysis.** We conducted an $F$-test using GLM 2, to test whether any of the shared features were significantly correlated with a given voxel (a diagonal with one at all features in SPM). The resulting F-map is thresholded at $p < 0.05$ cFWE at the whole-brain with height threshold at $p < 0.001$. We then asked within the survived voxels, which of them were also significantly positively correlated with PPI regressor in GLM 4, using at value thresholded at $p < 0.001$ uncorrected. We then counted the fraction of voxels that survived this test in a given ROI.

**Regression analysis with cross validation.** In addition to the SPM GLM analysis, we also performed regression analyses with cross validation within each participant[67]. We first extracted beta estimates at stimulus presentation time on each trial from a GLM with regressors at each stimulus onset, where the GLM also included other nuisance regressors, including framewise displacement, comp-cor, non-steady, trans, rot, the onsets of Decision, Action and feedback. We then used these beta estimates at the stimulus presentation time as dependent variables in our regression analysis. In all fMRI analyses, we used a Lasso penalty unless otherwise stated. The hyperparameters were optimized using 12-fold cross validation. The Matlab lasso function was used. We note that each stimulus was presented only once in our experiment in a given participant.

We performed a feature coding analysis analogous to what we performed using SPM. We first estimated the weights of the LFS model

features using lasso regression at each voxel. We then computed a sum of squared weights for low-level features and high-level features separately. In order to discard weight estimates that can be obtained by chance, we also performed the same lasso regression analysis using shuffled stimuli labels. We then constructed a null distribution with a sum of squared weights at each ROI using the weight estimates from this analysis. If the sum of squared weights of low (or high) -level features obtained from correct stimuli labels at a given voxel is significantly larger than the null distribution of low (or high) level features in the ROI ($p < 0.001$), we identified the voxel as encoding low-level (or high-level) features.

We also ran a similar analysis with the LFS model's features where we also included 'nonlinear features' that are constructed by multiplying pairs of the LFS model's features. As described above, we grouped the nonlinear features into three groups. (1) features created from interactions between high-level features (2) features created from interactions between low-level features and (3) features created from interactions between high-level and low-level features. We took five PCs from each group to match with the number of original high-level features from the model.

When comparing predictive accuracy across different models, we calculated Pearson correlations between the data and each model's predictions, where the model's predictions were pooled over predictions from testing sets across cross-validations.

We performed a similar analysis using the DCNN's features, where the DCNN was trained to predict behavioral data. Using the obtained results, we computed the sum of squared features from layers one to four, layers five to nine, layers ten to thirteen, and layers fourteen to fifteen. Again, estimates that are significantly greater than the ones obtained by chance (at $p < 0.001$) were included in our results, using the same regression analysis with shuffled labeled data. We performed analyses with 45 features (3 PCs from each layer) and 150 features (10 PCs from each layer).

We also performed the same DCNN analysis using untrained, random, weights.

**Reporting summary**
Further information on research design is available in the Nature Portfolio Reporting Summary linked to this article.

## Data availability
The data that support the findings of this study are available at https://github.com/kiigaya/Art or from the corresponding author upon request. Source data are provided with this paper.

## Code availability
The code that supports the findings of this study are available at https://github.com/kiigaya/Art or from the corresponding author upon request.

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

## Acknowledgements

We thank Peter Dayan, Shin Shimojo, Pietro Perona, Lesley Fellows, Avinash Vaidya and Jeff Cockburn for discussions and suggestions. We also thank Ronan O'Doherty for drawing the bird and fruit-bowl paintings, Seiji Iigaya and Erica Iigaya for drawing the color field painting presented in this manuscript. This work was supported by NIDA grant R01DA040011 and the Caltech Conte Center for Social Decision Making (P50MH094258) to J.O.D., the Japan Society for Promotion of Science, the Swartz Foundation and the Suntory Foundation to K.I., and the William H. and Helen Lang SURF Fellowship to I.A.W.

## Author contributions

K.I. and J.P.O. conceived and designed the project. K.I., S.Y., I.A.W., S.T., performed experiments and K.I., S.Y., I.A.W., L.C., J.P.O. analyzed and discussed results. K.I., S.Y., I.A.W., J.P.O. wrote the manuscript.

## Competing interests

The authors declare no competing interests.
