## [Peer Review File · Nature Communications]

Neural mechanisms underlying the hierarchical construction of perceived aesthetic valueREVIEWER COMMENTS

Reviewer #1 (Remarks to the Author):

The manuscript by ligaya et al. investigates the nature of how aesthetic value is computed in the human brain. A large fMRI dataset is collected in response to subjects perceiving and making subjective value judgments on visual artistic stimuli; this dataset emphasizes large amounts of fMRI data in a small number of subjects. Several computationally sophisticated analyses and models are brought to bear on interpreting the fMRI data. While many of the models and methods have been previously used to understand representation in visual cortex, the present study takes these ideas and approaches into the domain of art and aesthetics. This is an interesting novel direction. The authors consider two different kinds of image-computable models (one based on deep neural networks (DNN), one based on a simpler and most interpretable "feature-based" model), and map these models onto brain responses in different areas including visual, parietal, and frontal regions. The main novel findings concern value construction in frontal regions (whereas the results in visual regions are less novel).

There is much to like about the current manuscript, as it tackles an interesting topic (aesthetics) with appropriate computational methods, seeking to build general predictive models of value computation. The consideration of both the DNN and the feature-based model is interesting and provides some insights across the two types of models. In addition, the writing and general conceptual setup is clear and well done. However, while the computational and technical approaches used in the paper are sound at a coarse level, I find major reservations in how they are exactly conducted and reasoned about (elaborated on below). In addition, the theoretical framework of the present work is a bit shallow, at least in its current form, and so the ultimate strength of evidence that the current manuscript bears to bear on the nature of value computation in the brain is fairly limited.

MAJOR CONCERNS:

1. **STRENGTH OF THE MODELING EFFORTS.** The authors have already established in prior work that, from a purely behavioral perspective, image computable features (e.g. derived from a DNN) can predict a given subject's subjective aesthetic judgements. Thus, what's at stake in the present paper is how deeply can connections be made between these observations and the types of computations that might be happening in the brain. In its current form, the results demonstrated in the manuscript provide limited evidence. The paper feels a bit light on model alternatives and, to some degree, on the very modeling

premise (see Point 5). How exquisitely predictive are the current models for the brain data? If the models had extremely high predictive value (e.g. nearing the noise ceiling), that would be compelling, but the current model performance does not seem to be very high. To strengthen and address these sorts of concerns, the manuscript could perform a more detailed analysis of the quantitative performance of the models, could systematically entertain and adjudicate between different types of computational models, develop some more of the theoretical framework regarding the role of nonlinearities (Point 5), and so on.

2. MODELING AND REGRESSION PROCEDURES. The primary method that the manuscript uses to parcellate the relative influence of "low-level" and "high-level" image features is to use a F-test over features, binarize outcomes, and count proportions. This procedure seems a bit roundabout, a bit hard to interpret, and relies on arbitrary thresholds; and it is not clear that this is providing us a clear picture of the issues at stake. Perhaps this might be a limitation of what the SPM package enables. In any case, it would seem that adopting a more customized regression and modeling analysis (like the machine learning analyses performed elsewhere in the paper) would be helpful.

Such an analysis would proceed using basic regression procedures, such as carefully examining the regression weights estimated for different features (and their reliability), quantifying the variance that such features explain in different voxels and brain areas, considering noise ceilings (i.e. quantification of the intrinsic noise level in each voxel's data), and things like that. Regarding differences in SNR across regions (which the manuscript does recognize), this could be handled by comparing, for a given voxel, the relative variance contributions of the weights. It would also be informative to see a fuller treatment of how well and in exactly what way the computational model features map onto brain voxels.

There is a specific concern about the reduction of SNR in the frontal regions. Specifically, the paper asserts that a "feature breakdown" of 0.5/0.5 reflects that these ROIs encode things in a "mixed manner"; this is one of the core claims of the paper. But it might be that such a breakdown is the expectation of what would occur in voxels with high noise (which we can define here as having a signal that is poorly modeled by the currently entertained model features). It might be that the Type I error rate for the F-test might start to exhibit more even mixtures of false positive voxels favoring low-level vs. high-level features. In any case, careful consideration of null distributions (e.g. shuffling image-to-response assignments) and their corresponding outcomes would seem critical.

Note that these types of methodological concerns also extend to the DNN model. The use of 3 PCs to summarize the DNN features is well-intentioned certainly (to avoid overfitting) but ultimately arbitrary. It is suggested that the authors consider framing the mapping in terms of a principled form of regularized linear regression, such as ridge regression or the LASSO.

3. LOCALIZATION AND MAPPING. The current depiction of the results of brain mapping is not very systematic or comprehensive (e.g. Fig 3). While I understand that there is substantial neuroanatomical

variability across subjects (especially in frontal cortex), a more systematic depiction of results is necessary. One possibility (but not absolutely necessary) is to show cortical surfaces and/or flat maps, which have the advantage of seeing most or all of the brain in a single figure. Or, alternatively, showing all subjects in a common volume-based MNI space with an image montage or something like that might be helpful. Currently, the figures have the semblance of being "cherry picked" to show specific slices and it's not clear why different subjects are shown with different slices. Having a more comprehensive set of visualizations will help the reader to assess the level of evidence for the claims of the study.

MODERATE CONCERNS:

4. MODEL vs. BRAIN. One issue with the present manuscript has to do with how it ascribes the nature of the modeling that the authors used to the brain itself. That is, in many points in the paper, the paper claims that the brain performs certain operations. For example, consider the phrases "integrates the value", "features are then projected", "we show that input images are decomposed into hierarchically represented", "computed by summing over these features", and "projected into the PPC and PFC to form a rich feature space". But this conflates the model with the brain itself. In the context of a specific model, one can certainly state that the *model* performs such operations. However, it requires substantially more validation and evidence (e.g. consideration of alternative models) before one can claim that one's results justify the ascription of a model's operations to the brain itself. For example, consider the hypothetical case in which one uses a linear model to describe some brain data in terms of some generic predictors. This, in and of itself, does not entail that the brain is actually performing a weighted sum of those predictors.

I do not want to exaggerate the severity of this issue --- it can be easily resolved by more careful and measured phrasing and writing and language when discussing the modeling work. However, note that there is a deeper issue that this taps into. Just because a given model "works" to some degree on brain data (e.g. a hierarchical feature construction model seems to have some predictive value for frontal cortex responses), doesn't necessarily mean that the model is actually a good one. Thus, this leads to an open question of thinking about alternative models and whether we can rule them out (see Point 1).

5. NONLINEARITIES. While there are nonlinear computations embedded in the various computational models used in the manuscript, the techniques used to map the models to the brain are linear techniques. This raises some conundrums. For example, if voxels show linear relationships to the features in the LFS model, and if the features of the LFS model are sufficient for predicting subjective value, then shouldn't the value signals in frontal cortex also be linearly capturable by the LFS features? If

subjective value seems to be encoded in mPFC, and given that the authors have already shown that LFS features have linear predictive power for predictive subjective (behavioral) judgements in their prior work, is it a trivial consequence that the fMRI signals in mPFC can be modeled to some degree as a weighted combination of LFS features?

The larger point here is that the modeling framework that is being brought to bear on the fMRI data seems somewhat underdeveloped. Where are the nonlinearities? If the present manuscript does not quite know what the exact nature of the nonlinearities being computed in the brain are, can we at least get a handle how such nonlinearities are expected to manifest in the current set of results?

The authors allude to and recognize this issue, to some extent, on page 14 in the discussion of the correlations between value and visual features. I would recommend a further expansion and rethinking on these issues.

MINOR CONCERNS:

6. It is not clear what value the PPI analysis has or the conceptual point that it is making. Is the claim of the PPI analysis that certain regions tend to have correlated activity with one another?

7. It would be nice to have some sort of visual comparison between performance in the DNN and LFS models.

8. Figure 1G and its caption do not seem to be matched.

9. The text in the caption of Figure 2A is not clear enough to understand what the analysis was.

10. Please clarify what -30 degrees slice orientation means.

11. Was parallel acceleration used for the fMRI sequence?

12. Please provide T1 and T2 pulse sequence parameters.

13. Please clarify "regularization coefficient of the regression" on p. 24. Is this ridge regression?

14. Please clarify what the metric of model goodness of fit/prediction was for the decoding analysis (p. 24).

15. One (optional) control analysis that might be helpful for Figure 2 (and any DNN-derived results in general) would be to recompute both Figure 2a and Figure 2b for a version of the DNN with random weights (e.g., Kell [2018], Neuron). This would help tease apart to what extent the model's ability to predict the LFS features (or neural responses) reflects (1) task optimization of DNN weights, (2) the architecture of the DNN, (3) or random chance.

16. Although perhaps out of scope of the present manuscript, the authors may want to consider why different subjects have different levels of accuracy in the prediction of their subjective judgments. Are these interesting individual differences? Do they reflect different levels of behavioral reliability? Different neural mechanisms? More generally, this leads into issues like a deeper consideration of variance explained, noise ceilings, and things like that (see Point 2).

=====

Kendrick Kay, kay@umn.edu

Assistant Professor

University of Minnesota

*As of June 2017, I now sign all paper reviews. I am happy to clarify comments if they are not clear.

Reviewer #3 (Remarks to the Author):

I enjoyed reading this manuscript, and believe that it makes an important contribution not only to the domains of neuroaesthetics and empirical aesthetics, but also to neuroeconomics more broadly. As the authors have noted, because their value construction model proposed and tested here is agnostic to the

type of object that is being evaluated, it can potentially be applied to all sorts of other categories of object aside from visual art to which humans also attach value. Returning to artworks for a moment, the authors are correct in noting that currently we don't have a mechanistic explanation of how value for art is computed in the brain, and how that process leads to a subjective aesthetic rating (e.g., liking, beauty, etc.). All we know is that viewing artworks that are valued more is correlated with a relatively greater BOLD response in a set of regions in the brain. In that sense this manuscript represents a major advance in the field, and will likely also play an important role in placing neuroaesthetics more firmly within the larger reward literature. From my perspective, two findings in particular stand out here. First, using connectivity analysis with PPI, the results demonstrated that representations of high-level and low-level features are projected onto the posterior parietal (PPC) and lateral prefrontal cortices (LPFC), which are in turn utilized to compute a final aesthetic value in the medial prefrontal cortex (mPFC). Although further experimental work will be necessary to demonstrate the direction of the causal arrows in this connectivity model, there is now finally a plausible computational model that can be tested further with that level of precision. Second, from a theoretical perspective, it is intriguing that whereas low-level features are represented more robustly in early visual cortical areas, subjective value is represented more strongly in the PPC, LPFC and mPFC. A question of major interest in neuroaesthetics has been whether the computation of aesthetic value occurs already in sensory areas, or whether it occurs later in higher-order regions such as the OFC, etc. The present results add important new data regarding that question to the literature, to which I will return below. Nevertheless, I do have some questions and concerns that I hope the authors can address to improve the manuscript further.

1. I have some questions about how low- and high-level visual features are conceptualized here. Low-level features are pretty straightforward, and include variables such as colour distributions, brightness effects, etc. In turn, the three high-level features included concreteness, dynamics, and temperature. Is "concreteness" defined in terms of the abstract-representational dimension? Similarly, by "dynamics" do you mean perceived movement? I am not sure what is meant by "temperature" (Intensity? Saliency?). I realize that these terms might have been defined elsewhere, but it is important that they be defined in the body of the present manuscript as well. This is because as stated here, their referents would not be clear to researchers in empirical aesthetics. For example, Chatterjee et al. (2010) distinguished between two types of attributes: Formal-Perceptual (balance, color saturation, color temperature, depth, complexity, stroke) vs. Content-Representational (abstraction, animacy, emotion, realism, objective accuracy, and symbolism). I'm not sure how those attributes map onto the low- and high-level visual features described by the authors. Please explain.

2. Related to the point above, I also have some questions about how low- and high-level visual features are presumed to be related to each other. Specifically, the authors "hypothesized that high-level features are constructed from low-level features, and that subjective value is constructed from a linear combination of all low and high-level features." How and why are high-level features such as concreteness, dynamics, and temperature presumed to be constructed from low-level features. In most standard information-processing models of aesthetic experience (e.g., Leder et al., 2004, etc.), there is indeed bottom-up perceptual input into the system, but then they interact with top-down effects (e.g., context, expertise, etc.) to generate an aesthetic judgment. It is not typically assumed that the high-level

features arise out of the low-level features, because those two types of features are viewed as different types of inputs into the system. By low-level and high-level do you mean low and high levels of strictly perceptual input? Please clarify.

3. Although the researchers did not administer a standard instrument for assessing art experience, it appears that the participants had no formal training in the visual arts. It is likely that this had an impact on the results because art expertise is known to influence the way we view (based on eye-tracking data) and evaluate paintings. Aaron Kozbelt has several papers on this theme, including this recent summary chapter that has all the key references:

<https://www.oxfordhandbooks.com/view/10.1093/oxfordhb/9780198824350.001.0001/oxfordhb-9780198824350-e-37>

Of particular relevance here, Hekkert and van Wieringen (1996) and others have shown that features such as degree of realism (figurative vs abstract) and color (color vs black-and-white) have a much stronger impact on aesthetic preference in people with no formal training in the arts than amongst experts, who are in turn more in tune with the structural properties (e.g., compositional geometry) of artworks than non-experts. The gist of this research is that superficial features of artworks (e.g., colour) are more important drivers of aesthetic evaluation for people with no training in the visual arts than in experts, who are in turn more focused on compositional, structural and conceptual aspects of art.

Hekkert, P., & van Wieringen, P. C. W. (1996). The impact of level of expertise on the evaluation of original and altered versions of post-impressionistic paintings. *Acta Psychologica*, 94(2), 117–131. [https://doi.org/10.1016/0001-6918\(95\)00055-0](https://doi.org/10.1016/0001-6918(95)00055-0)

I think that it would be helpful to add some consideration of this important topic in the Discussion, and to highlight that the pattern of results observed here could be moderated by factors that are known to impact aesthetic evaluation, in particular formal training. Expertise has also been shown to influence the neural correlates of aesthetic preference (for review see Chatterjee & Vartanian, 2014).

Chatterjee, A., & Vartanian, O. (2014). Neuroaesthetics. *Trends in Cognitive Sciences*, 18, 370–375. <https://doi.org/10.1016/j.tics.2014.03.003>

4. As I mentioned above, researchers in neuroaesthetics have wondered whether the computation of aesthetic value occurs already in sensory areas, or whether it occurs later in higher-order regions such as the OFC, etc. For example, Chatterjee et al. (2009) found that a distributed set of regions in the visual and higher-order regions were activated when subjects explicitly judged attractiveness of faces.

Interestingly, in a separate run in the same study when they were not attending explicitly to attractiveness but rather were judging facial identity, a subset of the same regions nevertheless remained active. Results such as those and others have suggested that a set of regions in sensory and adjacent areas might respond automatically to beauty. However, if I understand correctly, the present results suggest that subjective value is not represented in sensory areas, but rather in the PPC, IPFC and mPFC. Although this single finding does not rule out the possibility that sensory areas could contribute to the computation of value, it would be useful if the authors addressed this important issue in the Discussion, which would likely be of great interest to researchers in neuroaesthetics.

Chatterjee, A., Thomas, A., Smith, S. E., & Aguirre, G. K. (2009). The neural response to facial attractiveness. *Neuropsychology*, 23, 135-143. <https://doi.org/10.1037/a0014430>

5. Toward the end of the Discussion the authors have devoted an entire paragraph highlighting the advantages of their design, which consisted of scanning a small number of subjects many times. This is contrasted with “classic group-based neuroimaging study in which results are obtained from the group average of many participants, where each participant performs short sessions, thus providing data with low signal to noise.” Surely the authors must be aware that there are also disadvantages associated with their design, especially if the focus had been on exploring individual differences. I think that it is necessary to discuss the pros and cons of any design so that a balanced argument is presented in that regard.

Oshin Vartanian

Dear Reviewers and Editor,

Thank you very much again for the time you spent reading and providing us with suggestions during these unprecedented times. We apologize for the delay in returning a revised manuscript, which was due to unexpected circumstances caused by the pandemic. We have now performed extensive additional analyses and revised the manuscript to fully address the reviewers' concerns.

Specifically, the revised version includes fourteen new figures: a **new Figure 5** (Lasso regression results with LFS features + nonlinear features at visual areas), a **new Figure S5** (subjective value signal presented in the common slices), a **new Figure S9** (Lasso regression results with LFS features at ventral-temporal and dorso-lateral visual streams), a **new Figure S10** (Model's predictive accuracy across ROIs, linear vs nonlinear feature models and DCNN model.), a **new Figure S11** (Lasso regression results with LFS features + nonlinear features at visual areas, PPC, PFC), a **new Figure S12** (behavioral weights estimation from LFS features + nonlinear features), **new Figure S13** (behavioral prediction of LFS features and nonlinear features), a **new Figure S14** (Lasso regression results with 45 Deep Convolutional Neural Network features at ventral-temporal and dorso-lateral visual streams), a **new Figure S15** (Lasso regression results with 150 Deep Convolutional Neural Network features at ventral-temporal and dorso-lateral visual streams), a **new Figure S16** (Lasso regression results with DCNN model with random weights at ventral-temporal and dorso-lateral visual streams), and a **new Figure S17** (Lasso regression results with DCNN model with random weights at PPC, PFC), a **new Figure S18** (Lasso regression results with LFS features at visual areas, posterior parietal cortex (PPC), lateral prefrontal cortex (IPFC), medial prefrontal cortex (mPFC)), a **new Figure S19** (Lasso regression results with 45 DCNN model's features at visual areas, PPC, IPFC, mPFC), a **new Figure S20** (Lasso regression results with 150 DCNN model's features at visual areas, PPC, IPFC, mPFC). Further, we have extensively edited our manuscript to address the reviewer's suggestions.

Here we outline our responses to the reviewers' comments in a point-by-point fashion. For ease of reference, the changes are highlighted below, with the reviewers' comments in bold followed by our response.

REVIEWER COMMENTS

Reviewer #1 (Remarks to the Author):

The manuscript by ligaya et al. investigates the nature of how aesthetic value is computed in the human brain. A large fMRI dataset is collected in response to subjects perceiving and making subjective value judgments on visual artistic stimuli; this dataset emphasizes large amounts of fMRI data in a small number of subjects. Several computationally sophisticated analyses and models are brought to bear on interpreting the fMRI data. While many of the models and methods have been previously used to understand representation in visual

cortex, the present study takes these ideas and approaches into the domain of art and aesthetics. This is an interesting novel direction. The authors consider two different kinds of image-computable models (one based on deep neural networks (DNN), one based on a simpler and most interpretable "feature-based" model), and map these models onto brain responses in different areas including visual, parietal, and frontal regions. The main novel findings concern value construction in frontal regions (whereas the results in visual regions are less novel).

There is much to like about the current manuscript, as it tackles an interesting topic (aesthetics) with appropriate computational methods, seeking to build general predictive models of value computation. The consideration of both the DNN and the feature-based model is interesting and provides some insights across the two types of models. In addition, the writing and general conceptual setup is clear and well done. However, while the computational and technical approaches used in the paper are sound at a coarse level, I find major reservations in how they are exactly conducted and reasoned about (elaborated on below). In addition, the theoretical framework of the present work is a bit shallow, at least in its current form, and so the ultimate strength of evidence that the current manuscript bears to bear on the nature of value computation in the brain is fairly limited.

We appreciate this reviewer's positive comments. We have now addressed the reviewer's concerns with extensive additional analyses and revisions (please see below).

MAJOR CONCERNS

1. STRENGTH OF THE MODELING EFFORTS. The authors have already established in prior work that, from a purely behavioral perspective, image computable features (e.g. derived from a DNN) can predict a given subject's subjective aesthetic judgements. Thus, what's at stake in the present paper is how deeply can connections be made between these observations and the types of computations that might be happening in the brain. In its current form, the results demonstrated in the manuscript provide limited evidence. The paper feels a bit light on model alternatives and, to some degree, on the very modeling premise (see Point 5). How exquisitely predictive are the current models for the brain data? If the models had extremely high predictive value (e.g. nearing the noise ceiling), that would be compelling, but the current model performance does not seem to be very high. To strengthen and address these sorts of concerns, the manuscript could perform a more detailed analysis of the quantitative performance of the models, could systematically entertain and adjudicate between different types of computational models, develop some more of the theoretical framework regarding the role of nonlinearities (Point 5), and so on.

In order to address the reviewer's concerns, we now performed all the original and new analyses using sparse linear regressions with cross validation (instead of using SPM). We performed additional model-driven analyses, including the testing of alternative models (including a model with nonlinear features and a variant of the DCNN model with random weights), while also comparing accuracies between models. We also performed a more detailed analysis of the quantitative performance of the models while developing more of the theoretical framework regarding the role of nonlinearities. We quantified significance against null distributions constructed from the same analyses with permuted label data. However, we note, that comparisons of model predictions with a noise ceiling are not feasible here, because we did not repeat multiple trials of the same type (the same stimuli were never repeated), which would have been necessary to allow direct estimation of a noise ceiling. We detail our responses to each of these requests in our response to the reviewer's comment 2 below, which goes into each of these concerns in more detail.

2. MODELING AND REGRESSION PROCEDURES. The primary method that the manuscript uses to parcellate the relative influence of "low-level" and "high-level" image features is to use a F-test over features, binarize outcomes, and count proportions. This procedure seems a bit roundabout, a bit hard to interpret, and relies on arbitrary thresholds; and it is not clear that this is providing us a clear picture of the issues at stake. Perhaps this might be a limitation of what the SPM package enables. In any case, it would seem that adopting a more customized regression and modeling analysis (like the machine learning analyses performed elsewhere in the paper) would be helpful. Such an analysis would proceed using basic regression procedures, such as carefully examining the regression weights estimated for different features (and their reliability), quantifying the variance that such features explain in different voxels and brain areas, considering noise ceilings (i.e. quantification of the intrinsic noise level in each voxel's data), and things like that. Regarding differences in SNR across regions (which the manuscript does recognize), this could be handled by comparing, for a given voxel, the relative variance contributions of the weights. It would also be informative to see a fuller treatment of how well and in exactly what way the computational model features map onto brain voxels. There is a specific concern about the reduction of SNR in the frontal regions. Specifically, the paper asserts that a "feature breakdown" of 0.5/0.5 reflects that these ROIs encode things in a "mixed manner"; this is one of the core claims of the paper. But it might be that such a breakdown is the expectation of what would occur in voxels with high noise (which we can define here as having a signal that is poorly modeled by the currently entertained model features). It might be that the Type I error rate for the F-test might start to exhibit more even mixtures of false positive voxels favoring low-level vs. high-level features. In any case, careful consideration of null distributions (e.g. shuffling image-to-response assignments) and their corresponding outcomes would seem critical. Note that these types of methodological concerns also extend to the DNN model. The use of 3 PCs to summarize the DNN features is well-intentioned certainly (to avoid overfitting) but

ultimately arbitrary. It is suggested that the authors consider framing the mapping in terms of a principled form of regularized linear regression, such as ridge regression or the LASSO.

We appreciate these comments. As we outlined in our response to point #1, following the reviewer's suggestion, we now also performed all analyses using lasso regression without using SPM. Instead of performing an F-test, we evaluated our results against null distributions that are constructed from our analysis with shuffled image labels as suggested by the reviewer. We also performed lasso regressions with larger sets of DCNN model features. In addition, we tested a new model with nonlinear features and a DCNN model with random weights. The details are as follows:

We re-analyzed our dataset using lasso regressions with cross validation in each participant.

(Page 10)

We also performed additional encoding analysis using cross validation at each voxel of each participant. Specifically, we performed a lasso regression at each voxel with the low- and high-level features that we considered in our original analyses. Hyperparameters are optimized in 12-fold cross validation at each voxel across stimuli.

(Method section) In addition to the SPM GLM analysis, we also performed regression analyses with cross validation within each participant. We first extracted beta estimates at stimulus presentations from each trial from GLM with regressors at each stimulus onset, where the GLM also includes other nuisance regressors, including framewise displacement, comp-cor, nonsteady, trans, rot, the onsets of Decision, Action and feedback. We then used these beta estimates at stimuli presentations as dependent variables in our regression analysis. In all fMRI analyses, we used Lasso penalty unless otherwise stated. The hyperparameters were optimized using 12-fold cross validation. Matlab function lasso was used. We note that each stimulus was presented only once in our experiment in a given participant.

We examined the weight structure at each voxel. The significance was compared against a null distribution constructed from the same analysis on shuffled labeled data.

(page 10) As a robustness check, we determined if our GLM results can be reproduced using the lasso regression analysis. We analyzed how low-level feature weights and high-level feature weights changed across ROIs. For this, we computed the sum of squares of low-level feature weights and the sum of squares of high-level feature weights at each voxel. Because these weights estimates include those that can be obtained by chance, we also computed the same quantities by performing the lasso regression with shuffled stimuli labels (labels

were shuffled at every regression). The null distribution of feature magnitudes (the sum of squares) was estimated for low-level features and high-level features at each ROI. For each voxel, we asked if estimated low-level features and high-level features are significantly larger than what is expected from noise, by comparing the magnitude of weights against the weights from null distribution ($p < 0.001$). We then examined how encoding of low-level vs high-level features varied across ROIs, as we did in our original GLM analysis.

(Method section) We performed a feature coding analysis analogous to what we performed using SPM. We first estimated the weights of LFS model's features using lasso regression at each voxel. We then computed a sum of squared weights for low-level features and high-level features separately. In order to discard weight estimates that can be obtained by chance, we also performed the same lasso regression analysis using shuffled stimuli labels. We then construct a null distribution with a sum of squared weights at each ROI using the weight estimates from this analysis. If the sum of squared weights of low (or high) -level features obtained from correct stimuli labels at a given voxel is significantly larger than the null distribution of low (or high) level features at the ROI ($p < 0.001$), we identified the voxel as encoding low-level (or high-level) features.

We found that this analysis largely reproduces what we have found in our original analysis.

(page 10) As seen in Figure S9, the original GLM analysis results were largely reproduced in the lasso regression. Namely, low-level features are more prominently encoded in early visual regions, while high-level features are more prominently encoded in higher visual regions. In this additional analysis such effects were clearly seen across five out of six participants, while one participant (P1) showed less clear early vs late region-specific differentiation with regard to low vs high-level feature representation. We also note that the model's predictive accuracy in visual regions was lower for this participant (P1) than for the rest of the participants (Figure 10).

Following the reviewer's suggestion (points 1, 2 and 5), we also tested a model with nonlinear features both on behavior and fMRI signals.

(Page 10) However, it is possible that nonlinear combinations of these features are also represented in the brain and that these may contribute to value computation. To explore this possibility, we constructed a new set of nonlinear features by multiplying pairs of the LFS model's features (interaction terms). We grouped these new features into three groups: interactions between pairs of low-level features (low-level x low-level), interactions between pairs of low-level and high-level features (low-level x high-level), and interactions between pairs of high-level features (high-level x high-level). To control the dimensionality of the new feature groups, we performed principal component analysis within each of the three

groups of non-linear features, and took the first five PCs to match the number of the high-level features specified in our original LFS model. We performed a LASSO regression analysis with these new features and the original features.

(Method section) We also ran a similar analysis with the LFS model's features where we also included 'nonlinear features' that are constructed by multiplying pairs of the LFS model's features. As described above, we grouped the nonlinear features into three groups. 1) features created from interactions between high level features 2) features created from interactions between low level features and 3) features created from interactions between high-level and low-level features. We took five PCs from each group to match with the number of original high-level features from the model.

When comparing predictive accuracy across different models, we calculated Pearson correlations between the data and each model's predictions, where the model's predictions were pooled over predictions from testing sets across cross-validations.

We found that nonlinear features constructed from pairs of high-level features significantly improved predictive accuracy for the fMRI signals, while they also predicted behavioral data.

(page 11) We found that in most participants, non-linear features created from pairs of high level features produced significant correlations with neural activity across multiple regions, while also showing similar evidence for a hierarchical organization from early to higher order regions, as found for the linear high level features (Figure 5, S11). Though comparisons between separately optimized lasso regressions should be cautiously interpreted, the mean correlations of the model with both linear and nonlinear features across ROIs showed a slight improvement in predictive accuracy compared to the original LFS model with only linear features (Figure S10), while the DCNN model's features out-performed both the original LFS model and the LFS model + nonlinear features.

Indeed, nonlinear features created from pairs of high-level features significantly contribute more to behavioral choice predictions than do other nonlinear features not built solely from high-level features (Figure S12). The first principal component of high level x high level features well captured three participants (3,5,6) behavior, while other participants show somewhat different weight profiles. However, we found that these newly added features only modestly improved the model's behavioral predictions (Figure S13).

(page 16) Going beyond our original LFS model, We also found that in most participants, non-linear features created from pairs of high level features specified in the original model produced significant correlations with neural activity across multiple regions, while largely showing similar evidence for a hierarchical

organization from early to higher order regions, as found for the linear high level features. These findings indicate that the brain encodes a much richer set of features than our original proposed set of low-level and high-level features as specified in the original LFS model. It will be interesting to see if the nonlinear features that we introduced here, especially the ones that were constructed from pairs of high-level features, can also be used to support behavioral judgments beyond the simple value judgments studied here, such as object recognition and other more complex judgements.

We also now performed a new analysis with LASSO regression using DCNN features, first with the same number of features as our previous analysis (45) and then with a much larger number of features (150). Again, we reproduced the original analysis results in ventral and dorsal pathways.

(Page 12) We also performed additional analyses with LASSO regression using the DCNN features. To test if we can reproduce the DCNN results originally performed with the GLM approach (as shown in Figure 4), we first performed LASSO regression with the same 45 features from all hidden layers. Hyperparameters were optimized by 12-fold cross-validation. The estimated weights were compared against the null distribution of each ROI constructed from the same analysis with shuffled stimuli labels. We then also performed the same analysis but with a larger set of features (150 features). In Figures S14, S15, we show how the weights on features from different layers varied across different ROIs in the visual stream. We computed the sum of squared weights of hidden layer groups (layer 1-4, 5-9, 10-13, 14-15). Again, in order to discard weight estimates that can be obtained by chance, we computed a null distribution by repeating the same analysis with shuffled labels and take the weight estimates that are significantly larger than the null distribution (at $p < 0.001$) in each ROI. We again found that LASSO regression with within-subject cross validation reproduced our original GLM analysis results..

(Methods section) We performed a similar analysis using the DCNN's features, where the DCNN was trained to predict behavioral data. Using the obtained results, we computed the sum of squared features from layers one to four, layers five to nine, layers ten to thirteen, and layers fourteen to fifteen. Again, estimates that are significantly greater than the ones obtained by chance (at $p < 0.001$) were included in our results, using the same regression analysis with shuffled labeled data. We performed analyses with 45 features (3 PCs from each layer) and 150 features (10 PCs from each layer).

Following the reviewer's suggestion (point 15), we also performed a lasso regression analysis with the DCNN's features but this time the model had random weights. Unlike the fully trained DCNN, this analysis shows no systematic structured correlation between the model's layers across ROIs.

(Page 12) As a control analysis, we asked whether similar results could be obtained from a DCNN model with random, untrained, weights.⁸⁵ We repeated the same LASSO regression analysis as we did in our analysis with the trained DCNN model. We found that such a model does not reproduce the finding of a hierarchical representation of layers that we found across the visual stream and other cortical areas that we found in the analysis with trained DCNN weights (Figures S16 S17).

(Methods section) We also performed the same DCNN analysis using untrained, random weights.

One limitation of our current experiment is that each stimulus was presented to each participant only once; this precludes calculating a noise ceiling directly. Liking judgments and fMRI signals are inherently noisy. Thus, a comparison of the model's predictive accuracy with the noise ceiling is something that will need to be pursued in subsequent research.

(page 18) Further, behavior and fMRI signals can be inherently noisy in that there will be a portion of data that cannot be predicted (i.e. a noise ceiling). Characterizing the contribution of these noise components will require further experiments with repeated measurements of decisions about the same stimuli.

3. LOCALIZATION AND MAPPING. The current depiction of the results of brain mapping is not very systematic or comprehensive (e.g. Fig 3). While I understand that there is substantial neuroanatomical variability across subjects (especially in frontal cortex), a more systematic depiction of results is necessary. One possibility (but not absolutely necessary) is to show cortical surfaces and/or flat maps, which have the advantage of seeing most or all of the brain in a single figure. Or, alternatively, showing all subjects in a common volume-based MNI space with an image montage or something like that might be helpful. Currently, the figures have the semblance of being "cherry picked" to show specific slices and it's not clear why different subjects are shown with different slices. Having a more comprehensive set of visualizations will help the reader to assess the level of evidence for the claims of the study.

We now have shown all subjects in a common MNI space with image montage (figure S5, 6). As you can see, the activations are heterogeneous but overlap. We took the cluster with the highest correlation as the ROI for value coding in each participant.

MODERATE CONCERNS:

4. MODEL vs. BRAIN. One issue with the present manuscript has to do with how it ascribes the nature of the modeling that the authors used to the brain itself. That is, in many points in the paper, the paper claims that the brain performs certain operations. For example, consider the phrases "integrates the value", "features are then projected", "we show that input images are decomposed into hierarchically represented", "computed by summing over these features", and "projected into the PPC and PFC to form a rich feature space". But this conflates the model with the brain itself. In the context of a specific model, one can certainly state that the *model* performs such operations. However, it requires substantially more validation and evidence (e.g. consideration of alternative models) before one can claim that one's results justify the ascription of a model's operations to the brain itself. For example, consider the hypothetical case in which one uses a linear model to describe some brain data in terms of some generic predictors. This, in and of itself, does not entail that the brain is actually performing a weighted sum of those predictors.

I do not want to exaggerate the severity of this issue --- it can be easily resolved by more careful and measured phrasing and writing and language when discussing the modeling work. However, note that there is a deeper issue that this taps into. Just because a given model "works" to some degree on brain data (e.g. a hierarchical feature construction model seems to have some predictive value for frontal cortex responses), doesn't necessarily mean that the model is actually a good one. Thus, this leads to an open question of thinking about alternative models and whether we can rule them out (see Point 1).

We now edited the manuscript to eliminate the misleading language. The reviewer is right that the fMRI results can only provide correlational evidence for the model's implementation. As noted earlier, we also now tested alternative models, including nonlinear features described above and a DCNN model with random weights. The fact that nonlinear features significantly correlate with the fMRI signal (especially the ones constructed from pairs of high level features) indicates that the brain is constructing a feature space that is richer than expected from our original simple LFS model. It is also notable that participant 1 was noisy and it was hard to see hierarchical structure in visual regions in the Lasso regression with the original model, but a hierarchical structure emerges more clearly even in that participant once we included nonlinear features (Figure 7, S21). We think it would be interesting to study whether these richer and more complex features representations might be used to support more complex behavioral judgements beyond the liking judgements we studied in the present study. We now added to the results section

(page 11) We found that in most participants, non-linear features created from pairs of high level features produced significant correlations with neural activity across multiple regions, while also showing similar evidence for a hierarchical

organization from early to higher order regions, as found for the linear high level features (Figure 5, S11). Though comparisons between separately optimized lasso regressions should be cautiously interpreted, the mean correlations of the model with both linear and nonlinear features across ROIs showed a slight improvement in predictive accuracy compared to the original LFS model with only linear features (Figure S10), while the DCNN model's features out-performed both the original LFS model and the LFS model + nonlinear features.

Indeed, nonlinear features created from pairs of high-level features significantly contribute more to behavioral choice predictions than do other nonlinear features not built solely from high-level features (Figure S12). The first principal component of high level x high level features well captured three participants (3,5,6) behavior, while other participants show somewhat different weight profiles. However, we found that these newly added features only modestly improved the model's behavioral predictions (Figure S13).

(page 16) Going beyond our original LFS model, We also found that in most participants, non-linear features created from pairs of high level features specified in the original model produced significant correlations with neural activity across multiple regions, while largely showing similar evidence for a hierarchical organization from early to higher order regions, as found for the linear high level features. These findings indicate that the brain encodes a much richer set of features than our original proposed set of low-level and high-level features as specified in the original LFS model. It will be interesting to see if the nonlinear features that we introduced here, especially the ones that were constructed from pairs of high-level features, can also be used to support behavioral judgments beyond the simple value judgments studied here, such as object recognition and other more complex judgements.

5. NONLINEARITIES. While there are nonlinear computations embedded in the various computational models used in the manuscript, the techniques used to map the models to the brain are linear techniques. This raises some conundrums. For example, if voxels show linear relationships to the features in the LFS model, and if the features of the LFS model are sufficient for predicting subjective value, then shouldn't the value signals in frontal cortex also be linearly capturable by the LFS features? If subjective value seems to be encoded in mPFC, and given that the authors have already shown that LFS features have linear predictive power for predictive subjective (behavioral) judgements in their prior work, is it a trivial consequence that the fMRI signals in mPFC can be modeled to some degree as a weighted combination of LFS features?

We agree with the reviewer that such a signal correlation would be a trivial result; however, this is not what we showed in our analysis. First, we showed unique correlates

to value and features by computing partial correlations (e.g., figure 6 and S15, S16, S17). Second, in our PPI analysis, all the features and value signals are regressed out. Thus what our PPI analyses show are noise correlations (instead of signal correlations) between voxels representing value and other voxels when participants make value judgements of stimuli. We then show that these PPI correlated voxels that are noise correlated value-coding voxels significantly overlap with feature-coding voxels in IPFC and PPC, but not in such voxels in visual areas.

(page 15) In a further validation of our earlier feature encoding analyses, we found that the pattern of hierarchical feature representation in visual regions was unaltered by the inclusion of ratings in the GLM (Figure S17).

(page 15) These results suggest that rich feature representations in the PPC and lateral PFC could potentially be leveraged to construct subjective values in mPFC. However, it is also possible that features represented in visual areas are directly used to construct subjective value in mPFC. To test this, we examined which of the voxels representing the LFS model features across the brain are coupled with voxels that represent subjective value in mPFC at the time when participants make decisions about the stimuli. A strong coupling would support the possibility that such feature representations are integrated at the time of decision-making in order to support a subjective value computation.

To test for this, we first performed a psychological-physiological interaction (PPI) analysis, examining which voxels are coupled with regions that represent subjective value when participants made decisions (Figure 7B and S24). We stress that this is not a trivial signal correlation, as in our PPI analysis all the value and feature signals are regressed out. Therefore the coupling is due to noise correlation between voxels. Then we asked how much of the feature-encoding voxels overlap with these PPI voxels. Specifically, we tested for the fraction of feature-encoding voxels that are also correlated with the PPI regressor across each ROI. Finding overlap between feature encoding voxels and PPI connectivity effects would be consistent with a role for these feature encoding representations in value construction. We found that the overlap was most prominent in the PPC and IPFC, while there was virtually no overlap in the visual areas at all (Figure 7C), consistent with the idea that features in the PPC and IPFC, instead of visual areas, are involved in constructing subjective value representations in mPFC. A more detailed decomposition of the PFC ROI from the same analysis shows the contribution of individual sub-regions of lateral and medial PFC (Figure S25).

In fact, if we apply the same analysis during inter-trial-intervals, these noise correlations between value-voxels in mPFC and feature-coding voxels in PPC/IPFC disappear, showing a specificity that is predicted from our value-computation model.

(page 13) We also performed a control analysis to test the specificity of the coupling to an experimental epoch by constructing a similar PPI regressor locked to the epoch of inter-trial- intervals (ITIs). This analysis showed a dramatically altered coupling that did not involve the same PPC and PFC regions (Figure S26). These findings indicate that coupling between PPC and LPFC with mPFC value representations occurs specifically at the time that subjective value computations are being performed, suggesting that these regions are playing an integrative role of feature representations at the time of valuation. We however note that all of our analyses are based on correlations, which do not provide information about the direction of the coupling.

The larger point here is that the modeling framework that is being brought to bear on the fMRI data seems somewhat underdeveloped. Where are the nonlinearities? If the present manuscript does not quite know what the exact nature of the nonlinearities being computed in the brain are, can we at least get a handle how such nonlinearities are expected to manifest in the current set of results?

The authors allude to and recognize this issue, to some extent, on page 14 in the discussion of the correlations between value and visual features. I would recommend a further expansion and rethinking on these issues.

We now performed additional analyses with nonlinear features. We indeed found that features constructed from high level features show significant correlations across ROIs. Further, those nonlinear features also show significant behavioral correlations. Our results suggest that the brain used much richer feature representations than our current model.

(page 11) We found that in most participants, non-linear features created from pairs of high level features produced significant correlations with neural activity across multiple regions, while also showing similar evidence for a hierarchical organization from early to higher order regions, as found for the linear high level features (Figure 5, S11). Though comparisons between separately optimized lasso regressions should be cautiously interpreted, the mean correlations of the model with both linear and nonlinear features across ROIs showed a slight improvement in predictive accuracy compared to the original LFS model with only linear features (Figure S10), while the DCNN model's features out-performed both the original LFS model and the LFS model + nonlinear features.

Indeed, nonlinear features created from pairs of high-level features significantly contribute more to behavioral choice predictions than do other nonlinear features not built solely from high-level features (Figure S12). The first principal component of high level x high level features well captured three participants (3,5,6) behavior, while other participants show somewhat different weight profiles. However, we found that these newly added features only modestly improved the model's behavioral predictions (Figure S13).

(page 16) Going beyond our original LFS model, We also found that in most participants, non-linear features created from pairs of high level features specified in the original model produced significant correlations with neural activity across multiple regions, while largely showing similar evidence for a hierarchical organization from early to higher order regions, as found for the linear high level features. These findings indicate that the brain encodes a much richer set of features than our original proposed set of low-level and high-level features as specified in the original LFS model. It will be interesting to see if the nonlinear features that we introduced here, especially the ones that were constructed from pairs of high-level features, can also be used to support behavioral judgments beyond the simple value judgments studied here, such as object recognition and other more complex judgements.

MINOR CONCERNS:

6. It is not clear what value the PPI analysis has or the conceptual point that it is making. Is the claim of the PPI analysis that certain regions tend to have correlated activity with one another?

The PPI analysis is very important evidence supporting the model's neural implementation. It shows that voxels that encode our model's features in IPFC and PPC are functionally coupled with voxels that encode aesthetic value in mPFC. However features encoded in visual areas did not show coupling with the value coding voxels in mPFC. This is evidence supporting that our model's feature-based value computation takes place between PPC/IPFC and mPFC. We note that this coupling is unique when participants evaluate stimuli. Our control analysis focusing on inter-trial-intervals shows dramatically different coupling. We also note that our results are not driven by trivial signal correlations but noise correlations.

(page 15) These results suggest that rich feature representations in the PPC and lateral PFC could potentially be leveraged to construct subjective values in mPFC. However, it is also possible that features represented in visual areas are directly used to construct subjective value in mPFC. To test this, we examined which of the voxels representing the LFS model features across the brain are coupled with voxels that represent subjective value in mPFC at the time when participants make decisions about the stimuli. A strong coupling would support the possibility that such feature representations are integrated at the time of decision-making in order to support a subjective value computation.

To test for this, we first performed a psychological-physiological interaction (PPI) analysis, examining which voxels are coupled with regions that represent subjective value when participants made decisions (Figure 7B and S24). We

stress that this is not a trivial signal correlation, as in our PPI analysis all the value and feature signals are regressed out. Therefore the coupling is due to noise correlation between voxels. Then we asked how much of the feature-encoding voxels overlap with these PPI voxels. Specifically, we tested for the fraction of feature-encoding voxels that are also correlated with the PPI regressor across each ROI. Finding overlap between feature encoding voxels and PPI connectivity effects would be consistent with a role for these feature encoding representations in value construction. We found that the overlap was most prominent in the PPC and IPFC, while there was virtually no overlap in the visual areas at all (Figure 7C), consistent with the idea that features in the PPC and IPFC, instead of visual areas, are involved in constructing subjective value representations in mPFC. More detailed decomposition of the PFC ROI from the same analysis shows the contribution of individual sub-regions of lateral and medial PFC (Figure S25).

7. It would be nice to have some sort of visual comparison between performance in the DNN and LFS models.

We performed additional analysis to compare across models. Though one needs to be cautious how to interpret fitting results from separately optimized lasso regressions, correlations between the model's prediction and the fMRI signal across ROIs and participants shows that the DCNN's hidden layers model best describes the fMRI data.

(page 11) Though comparisons between separately optimized lasso regressions should be cautiously interpreted, the mean correlations of the model with both linear and nonlinear features across ROIs showed a slight improvement in predictive accuracy compared to the original LFS model with only linear features (Figure S10), while the DCNN model's features out-performed both the original LFS model and the LFS model + nonlinear features..

8. Figure 1G and its caption do not seem to be matched.

Thanks very much. We edited the caption.

9. The text in the caption of Figure 2A is not clear enough to understand what the analysis was.

We expanded the explanation in the caption.

10. Please clarify what -30 degrees slice orientation means.

It means that slices were tilted -30 degrees with respect to AC-PC line. We changed the wording.

11. Was parallel acceleration used for the fMRI sequence?

In our functional imaging, the sequence used in-plane acceleration factor 2, multiband slice acceleration factor 4. We added this in our method section.

12. Please provide T1 and T2 pulse sequence parameters.

We now added the parameters in the methods section.

13. Please clarify "regularization coefficient of the regression" on p. 24. Is this ridge regression?

It is a hyperparameter of the ridge regression. We edited the text.

14. Please clarify what the metric of model goodness of fit/prediction was for the decoding analysis (p. 24).

For the continuous features (e.g., rating, mean hue), Pearson correlation between the model's prediction and data were used as the metric for goodness of fit, while for the categorical features (e.g., presence of person), we calculated accuracy, area under curve (AUC), and F1 scores. All of these three metrics gave us almost identical slopes in the decoding vs layer plots. We now added this in the methods section.

15. One (optional) control analysis that might be helpful for Figure 2 (and any DNN-derived results in general) would be to recompute both Figure 2a and Figure 2b for a version of the DNN with random weights (e.g., Kell [2018], Neuron). This would help tease apart to what extent the model's ability to predict the LFS features (or neural responses) reflects (1) task optimization of DNN weights, (2) the architecture of the DNN, (3) or random chance.

We appreciate this suggestion. As we outlined in above points, we ran additional analyses with DCNN with random weights, finding that random DCNN hidden layers do not capture hierarchical nature of feature coding in visual stream.

(Page 12) As a control analysis, we asked whether similar results could be obtained from a DCNN model with random, untrained, weights.⁸⁵ We repeated the same LASSO regression analysis as we did in our analysis with the trained DCNN model. We found that such a model does not reproduce the finding of a hierarchical representation of layers that we found across the visual stream and

other cortical areas that we found in the analysis with trained DCNN weights (Figures S16 S17).

(Methods section) We also performed the same DCNN analysis using untrained, random weights.

16. Although perhaps out of scope of the present manuscript, the authors may want to consider why different subjects have different levels of accuracy in the prediction of their subjective judgments. Are these interesting individual differences? Do they reflect different levels of behavioral reliability? Different neural mechanisms? More generally, this leads into issues like a deeper consideration of variance explained, noise ceilings, and things like that (see Point 2).

We appreciate this comment. The reviewer is right that analyzing individual differences is out of the scope of our current manuscript with six participants. However, we could speculate on some reasons why we see variations in behavioral predictions. Our model is likely to miss meaningful features, such as personal experience associated with stimuli, which may involve regions such as hippocampus. As reviewers pointed out, behavioral ratings may also be noisy, though measuring behavioral reliability requires further experiments repeating the same stimuli multiple times (in the current design we show each stimulus once). We agree that this is an interesting point that can be addressed in the future with a modified experimental design. We discussed this in the discussion section.

(page 18) It is likely that some participants used features that our model did not consider, such as personal experience associated with stimuli. Brain regions such as the hippocampus may potentially be involved in such additional feature computations. Further, behavior and fMRI signals can be inherently noisy in that there will be a portion of data that cannot be predicted (i.e. a noise ceiling). Characterizing the contribution of these noise components will require further experiments with repeated measurements of decisions about the same stimuli.

We once again thank this reviewer's enthusiasm and suggestions.

Reviewer #3 (Remarks to the Author):

I enjoyed reading this manuscript, and believe that it makes an important contribution not only to the domains of neuroaesthetics and empirical aesthetics, but also to neuroeconomics more broadly. As the authors have noted, because

their value construction model proposed and tested here is agnostic to the type of object that is being evaluated, it can potentially be applied to all sorts of other categories of object aside from visual art to which humans also attach value. Returning to artworks for a moment, the authors are correct in noting that currently we don't have a mechanistic explanation of how value for art is computed in the brain, and how that process leads to a subjective aesthetic rating (e.g., liking, beauty, etc.). All we know is that viewing artworks that are valued more is correlated with a relatively greater BOLD response in a set of regions in the brain. In that sense this manuscript represents a major advance in the field, and will likely also play an important role in placing neuroaesthetics more firmly within the larger reward literature. From my perspective, two findings in particular stand out here. First, using connectivity analysis with PPI, the results demonstrated that representations of high-level and low-level features are projected onto the posterior parietal (PPC) and lateral prefrontal cortices (IPFC), which are in turn utilized to compute a final aesthetic value in the medial prefrontal cortex (mPFC). Although further experimental work will be necessary to demonstrate the direction of the causal arrows in this connectivity model, there is now finally a plausible computational model that can be tested further with that level of precision. Second, from a theoretical perspective, it is intriguing that whereas low-level features are represented more robustly in early visual cortical areas, subjective value is represented more strongly in the PPC, IPFC and mPFC. A question of major interest in neuroaesthetics has been whether the computation of aesthetic value occurs already in sensory areas, or whether it occurs later in higher-order regions such as the OFC, etc. The present results add important new data regarding that question to the literature, to which I will return below. Nevertheless, I do have some questions and concerns that I hope the authors can address to improve the manuscript further.

We appreciate the reviewer's very positive comments.

1. I have some questions about how low- and high-level visual features are conceptualized here. Low-level features are pretty straightforward, and include variables such as colour distributions, brightness effects, etc. In turn, the three high-level features included concreteness, dynamics, and temperature. Is "concreteness" defined in terms of the abstract-representational dimension? Similarly, by "dynamics" do you mean perceived movement? I am not sure what is meant by "temperature" (Intensity? Saliency?). I realize that these terms might have been defined elsewhere, but it is important that they be defined in the body of the present manuscript as well. This is because as stated here, their referents would not be clear to researchers in empirical aesthetics. For example, Chatterjee et al. (2010) distinguished between two types of attributes: Formal-Perceptual (balance, color saturation, color temperature, depth, complexity, stroke) vs. Content-Representational

(abstraction, animacy, emotion, realism, objective accuracy, and symbolism). I'm not sure how those attributes map onto the low- and high-level visual features described by the authors. Please explain.

We appreciate this comment. Our features actually originated from Chatterjee et al (2010). Vaidya et al (2017) performed a dimensionality reduction of the features in Chatterjee et al and identified three features, which we adapted as high-level features in our model. We call them high level, because these are annotated by human participants in our previous study (Iigaya et al. (2021)). We now clarified this in the text:

(Methods section) In,51 Chatterjee et al. pioneered this by introducing 12 features (color temperature, depth, abstract, realism, balance, accuracy, stroke, animacy, emotion, color saturation, complexity) that were annotated by human participants for 24 paintings, in which the authors have found that annotations were consistent across participants, regardless of their artistic experience. Vaidya et al.43 further collected annotations of these feature sets from artistically experienced participants for an additional 175 paintings and performed a principal component analysis, finding three major components that summarize the variance of the original 12 features. Inspired by the three principal components, we introduced three high-level features: concreteness, dynamics, and temperature.

2. Related to the point above, I also have some questions about how low- and high-level visual features are presumed to be related to each other. Specifically, the authors “hypothesized that high-level features are constructed from low-level features, and that subjective value is constructed from a linear combination of all low and high-level features.” How and why are high-level features such as concreteness, dynamics, and temperature presumed to be constructed from low-level features. In most standard information-processing models of aesthetic experience (e.g., Leder et al., 2004, etc.), there is indeed bottom-up perceptual input into the system, but then they interact with top-down effects (e.g., context, expertise, etc.) to generate an aesthetic judgment. It is not typically assumed that the high-level features arise out of the low-level features, because those two types of features are viewed as different types of inputs into the system. By low-level and high-level do you mean low and high levels of strictly perceptual input? Please clarify.

Our hypothesis was based on visual processing in the brain. The low-level features are constructed from simple computer vision algorithms, such as saturation and contrasts. We hypothesized that these features are constructed in early visual processing in the brain. The high-level features are annotated by human participants, and are, we presumed, more complex than the low-level features. Inspired by the architecture of the visual stream, as well as by other findings comparing visual processing with deep convolutional networks, we hypothesized that the brain first constructs low-level features and then builds high-level features in later areas. We tested this hypothesis in

multiple ways, and found strong evidence for this. We agree with the reviewer that our model does not include variables such as contexts and expertise. We now discussed this in the manuscript.

(Page 18) For instance, there is evidence that art experts tend to evaluate art differently from people with no artistic training.^{87, 88} It would be interesting to study if feature representations may differ between experts and non-experts, while probing whether the computational motif that we found here (hierarchical visual feature representation in visual areas, value construction in PPC and PFC) might be conserved across different levels of expertise. We should also note that the model's predictive accuracy about liking ratings varied across participants. It is likely that some participants used features that our model did not consider, such as personal experience associated with stimuli. Brain regions such as the hippocampus may potentially be involved in such additional feature computations.

3. Although the researchers did not administer a standard instrument for assessing art experience, it appears that the participants had no formal training in the visual arts. It is likely that this had an impact on the results because art expertise is known to influence the way we view (based on eye-tracking data) and evaluate paintings. Aaron Kozbelt has several papers on this theme, including this recent summary chapter that has all the key references:

<https://www.oxfordhandbooks.com/view/10.1093/oxfordhb/9780198824350.001.0001/oxfordhb-9780198824350-e-37>

Of particular relevance here, Hekkert and van Wieringen (1996) and others have shown that features such as degree of realism (figurative vs abstract) and color (color vs black-and-white) have a much stronger impact on aesthetic preference in people with no formal training in the arts than amongst experts, who are in turn more in tune with the structural properties (e.g., compositional geometry) of artworks than non-experts. The gist of this research is that superficial features of artworks (e.g., colour) are more important drivers of aesthetic evaluation for people with no training in the visual arts than in experts, who are in turn more focused on compositional, structural and conceptual aspects of art.

Hekkert, P., & van Wieringen, P. C. W. (1996). The impact of level of expertise on the evaluation of original and altered versions of post-impressionistic paintings. *Acta Psychologica*, 94(2), 117–131. [https://doi.org/10.1016/0001-6918\(95\)00055-0](https://doi.org/10.1016/0001-6918(95)00055-0)

I think that it would be helpful to add some consideration of this important topic in the Discussion, and to highlight that the pattern of results observed here could be moderated by factors that are known to impact aesthetic evaluation, in particular formal training. Expertise has also been shown to influence the neural correlates of aesthetic preference (for review see Chatterjee & Vartanian, 2014).

Chatterjee, A., & Vartanian, O. (2014). Neuroaesthetics. *Trends in Cognitive Sciences*, 18, 370–375. <https://doi.org/10.1016/j.tics.2014.03.003>

We appreciate this comment. We now have cited these papers and we agree that studying different value computations between art experts and non-experts is an exciting topic. We added this in discussion.

(Page 18) While we found that results from the visual cortex were largely consistent across participants, the proportion of features represented in PCC and PFC, as well as the features that were used, were quite different across participants. Understanding such individual differences will be important in future work. For instance, there is evidence that art experts tend to evaluate art differently from people with no artistic training.^{87, 88} It would be interesting to study if feature representations may differ between experts and non-experts, while probing whether the computational motif that we found here (hierarchical visual feature representation in visual areas, value construction in PPC and PFC) might be conserved across different levels of expertise. We should also note that the model's predictive accuracy about liking ratings varied across participants. It is likely that some participants used features that our model did not consider, such as personal experience associated with stimuli. Brain regions such as the hippocampus may potentially be involved in such additional feature computations. Further, behavior and fMRI signals can be inherently noisy in that there will be a portion of data that cannot be predicted (i.e. a noise ceiling). Characterizing the contribution of these noise components will require further experiments with repeated measurements of decisions about the same stimuli.

4. As I mentioned above, researchers in neuroaesthetics have wondered whether the computation of aesthetic value occurs already in sensory areas, or whether it occurs later in higher-order regions such as the OFC, etc. For example, Chatterjee et al. (2009) found that a distributed set of regions in the visual and higher-order regions were activated when subjects explicitly judged attractiveness of faces. Interestingly, in a separate run in the same study when they were not attending explicitly to attractiveness but rather were judging facial identity, a subset of the same regions nevertheless remained active. Results such as those and others have suggested that a set of regions in sensory and adjacent areas might respond automatically to beauty. However, if I understand correctly, the present results suggest that subjective value is not represented in sensory areas, but rather in the PPC, IPFC and mPFC. Although this single finding does not rule out the possibility that sensory areas could contribute to the computation of value, it would be useful if the authors addressed this important issue in the Discussion, which would likely be of great interest to researchers in neuroaesthetics.

Chatterjee, A., Thomas, A., Smith, S. E., & Aguirre, G. K. (2009). The neural response to facial attractiveness. *Neuropsychology*, 23, 135-

143. <https://doi.org/10.1037/a0014430>

We appreciate this comment. Because the model's features are linearly correlated with aesthetic value, all of the voxels that show correlations with features are also correlated with subjective value, including visual areas. Thus, simple regression analysis with liking ratings shows correlations across many brain regions. However, in this manuscript we show that some of the subjective value correlates reflect features, rather than value. This is clear from our multiple regression analysis (Figure S15,16), where most correlations are loaded onto features rather than value. Thus our study suggests an additional possible interpretation for the findings by Chatterjee et al. (2009), which is that aesthetic value correlates in visual regions might be at least partly explained by visual features that correlate with aesthetic value. We discussed this important point in Discussion:

(page 17) Accumulating evidence has suggested that value signals can be found widely across the brain including even in sensory regions (e.g., 37, 57–62), posing a question about the differential contribution of different brain regions if value representations are so ubiquitous. While we also saw multiple brain regions that appeared to correlate with value signals during aesthetic valuation, our results suggest an alternative account for the widespread prevalence of value signals, which is that some components of the value signals especially in sensory cortex might reflect features that are ultimately used to construct value in later stages of information processing, instead of the value itself. Because neural correlates of features have not been probed previously, our results suggest that it may be possible to reinterpret at least some apparent value representations as reflecting the encoding of precursor features instead of value per se. In the present case even after taking into account feature representations, value signals were still detectable in the medial prefrontal cortex and elsewhere, supporting the notion that some brain regions are especially involved in value coding more than others. In future work it may be possible to even more clearly dissociate value from its sensory precursors by manipulating the context in which stimuli are presented, wherein features remain invariant across contexts, while the value changes. In doing so, further studies can illuminate finer dissociations between features and value signals.

5. Toward the end of the Discussion the authors have devoted an entire paragraph highlighting the advantages of their design, which consisted of scanning a small number of subjects many times. This is contrasted with “classic group-based neuroimaging study in which results are obtained from the group average of many participants, where each participant performs short sessions, thus providing data with low signal to noise.” Surely the authors must be aware that there are also disadvantages associated with their design, especially if the focus had been on exploring individual differences. I think that it is necessary to

discuss the pros and cons of any design so that a balanced argument is presented in that regard.

We agree with the reviewer. We now added this in the discussion.

(page 18) We note that one important limitation of this in-depth fMRI method is that it is not ideal for studying and characterizing differences across individuals. To gain a comprehensive account of such variability across individuals, it would be necessary to collect data from a much larger cohort of participants. As it is not feasible to scale the in-depth approach to such large cohorts due to experimenter time and resource constraints, such individual difference studies would necessarily require adopting more standard group-level scanning approaches and analyses.

REVIEWERS' COMMENTS

Reviewer #1 (Remarks to the Author):

The authors have performed a thorough and responsive revision. The additional modeling analyses/controls are helpful and improve the sophistication of the paper. The added text/discussion is also helpful in this regard.

I would suggest the authors add a brief remark regarding the fact that the interactions in the nonlinear model are just one way in which nonlinearities can be added on top of a linear model.

(Also, I would comment to the authors that coupling a PCA dimensionality reduction to a Lasso regression has some interactions --- what is sparse in the original space is no longer sparse in the PCA space, and vice versa --- and this creates some complexities for the interpretation of the regression. However, this analysis is ultimately just a minor detail in the big-picture of the paper.)

Finally, the manuscript needs some proof-reading and typo-checking. I noticed a few errors, but there may be more:

- p. 16 "We" -> "we"

- Fig S10 "Peason" -> "Pearson"

- p. 21 "0.8n" -> "0.8"?

Kendrick Kay

University of Minnesota

*As of June 2017, I now sign all paper reviews. I am happy to clarify comments if they are not clear.

Reviewer #2 (Remarks to the Author):

I would like to thank the authors for their careful and attentive handling of my suggestions for improving their manuscript. Building on its original strengths, the manuscript now does a much better job of acknowledging the interplay between bottom-up and top-down processes in the emergence of aesthetic

value/preference, including contextual factors within which it resides. I believe that this will be an important contribution to the literatures of empirical aesthetics and neuroaesthetics, and will introduce researchers in the field to novel approaches to measuring phenomena of interest.

Dear Reviewers

Thank you very much for your positive and helpful comments. Here is our response.

REVIEWERS' COMMENTS

Reviewer #1 (Remarks to the Author):

The authors have performed a thorough and responsive revision. The additional modeling analyses/controls are helpful and improve the sophistication of the paper. The added text/discussion is also helpful in this regard.

I would suggest the authors add a brief remark regarding the fact that the interactions in the nonlinear model are just one way in which nonlinearities can be added on top of a linear model.

We appreciate this comment. We added the following to the Discussion:

We also note that there are other ways to construct nonlinear features. Further studies with richer set of features, e.g, other forms of interactions, may improve behavioral and neural predictions.

(Also, I would comment to the authors that coupling a PCA dimensionality reduction to a Lasso regression has some interactions --- what is sparse in the original space is no longer sparse in the PCA space, and vice versa --- and this creates some complexities for the interpretation of the regression. However, this analysis is ultimately just a minor detail in the big-picture of the paper.)

Thank you very much for this comment. We will keep this mind in our future analyses.

Finally, the manuscript needs some proof-reading and typo-checking. I noticed a few errors, but there may be more:

- p. 16 "We" -> "we"*
- Fig S10 "Peason" -> "Pearson"*
- p. 21 "0.8n" -> "0.8"?*

We corrected these.

*Kendrick Kay
University of Minnesota*

**As of June 2017, I now sign all paper reviews. I am happy to clarify comments if they are not clear.*

Reviewer #2 (Remarks to the Author):

I would like to thank the authors for their careful and attentive handling of my suggestions for improving their manuscript. Building on its original strengths, the manuscript now does a much better job of acknowledging the interplay between bottom-up and top-down processes in the emergence of aesthetic value/preference, including contextual factors within which it resides. I believe that this will be an important contribution to the literatures of empirical aesthetics and neuroaesthetics, and will introduce researchers in the field to novel approaches to measuring phenomena of interest.

Thank you very much.